# **1** Atmospheric deposition fluxes over the Atlantic Ocean: A

# 2 **GEOTRACES case study**

- Jan-Lukas Menzel Barraqueta<sup>1,2</sup>, Jessica K. Klar<sup>3,4</sup>, Martha Gledhill<sup>1</sup>, Christian Schlosser<sup>1</sup>, Rachel
- Shelley<sup>5,6,7</sup>, Helene Planquette<sup>6</sup>, Bernhard Wenzel<sup>1</sup>, Geraldine Sarthou<sup>6</sup>, Eric P. Achterberg<sup>1</sup>
- <sup>1</sup> GEOMAR, Helmholtz Centre for Ocean Research Kiel, Germany
- <sup>2</sup> Department of Earth Sciences, Stellenbosch University, Stellenbosch, 7600, South Africa
- <sup>3</sup> Ocean and Earth Science, National Oceanography Centre, University of Southampton, European
- Way, Southampton SO14 3ZH, UK
- <sup>4</sup> LEGOS, Université de Toulouse, CNES, CNRS, IRD, UPS, 14 Avenue Edouard Belin, 31400 Toulouse,
   France
- <sup>5</sup> Department of Earth, Ocean and Atmospheric Science, Florida State University, 117 N Woodward
   Ave, Tallahassee, Florida, 32301, USA
- <sup>6</sup>Laboratoire des Sciences de l'Environnement Marin, UMR 6539 LEMAR
- (CNRS/UBO/IRD/IFREMER), Institut Universitaire Européen de la Mer, Technopôle Brest-Iroise,
- Plouzané 29280, LEMAR, UMR 6539, Plouzané, France
- <sup>7</sup> School of Geography, Earth and Environmental Sciences, University of Plymouth, Drake Circus,
- Plymouth, PL4 8AA, UK

# 18 Abstract

- Atmospheric deposition is an important source of micronutrients to the ocean, but atmospheric deposition
- fluxes remain poorly constrained in most ocean regions due to the limited number of field observations of wet
- and dry atmospheric inputs. Here we present the distribution of dissolved aluminium (dAl), as a tracer of
- atmospheric inputs, in surface waters of the Atlantic Ocean along GEOTRACES sections GA01, GA06, GA08, and
- GA10. We used the surface mixed layer concentrations of dAl to calculate atmospheric deposition fluxes using
- a simple steady state model. We have optimized the Al fractional aerosol solubility, dAl residence time within
- the surface mixed layer and depth of the surface mixed layer for each separate cruise to calculate the
- atmospheric deposition fluxes. We calculated the lowest deposition fluxes of  $0.15 \pm 0.1$  and  $0.27 \pm 0.13$  g m<sup>-2</sup>
- 27 yr<sup>-1</sup> for the South and North Atlantic Ocean (> 40°S and > 40°N), respectively, and highest fluxes of 1.8 and 3.09
- 28 g m<sup>-2</sup> yr<sup>-1</sup> for the South East Atlantic and tropical Atlantic Ocean, respectively. Overall, our estimations are
- comparable to atmospheric dust deposition model estimates and reported field-based atmospheric deposition
- estimates. We note that our estimates diverge from atmospheric dust deposition model flux estimates in
- regions influenced by riverine Al inputs and in upwelling regions. As dAl is a key trace element in the
- GEOTRACES Programme, the approach presented in this study allows calculations of atmospheric deposition
- fluxes at high spatial resolution for remote ocean regions.

### 34 Introduction

Atmospheric deposition is a major source of micronutrients, especially iron, to the surface ocean

(Martin et al., 1991; Moore et al., 2004). Aerosol deposition of iron in the tropical and subtropical

North Atlantic stimulates N<sub>2</sub> fixation (Moore et al., 2009), and in high latitude waters with high

nitrate low chlorophyll conditions it can enhance primary productivity (Baker et al., 2013).

Consequently, atmospheric deposition is considered to support up to 50 % of global export

production (Jickells et al., 2014). Therefore, by supplying growth-limiting elements to marine micro-

organisms, atmospheric deposition can have important direct impacts on the marine carbon cycle

and indirectly influence global climate (Mahowald et al., 2014).

The size distribution of mineral dust is considered a continuum. However, mineral dust is often 44 described to have a bimodal size distribution and thus aerosols are classified into a fine (radius 0.1-45 0.25  $\mu$ M) or coarse mode (radius 1 – 2.5  $\mu$ M) (Maring et al., 2003). The classification facilitates us to 46 assign aerosol deposition velocities for the aerosol size classes for the quantification of dry 47 deposition fluxes (Slinn and Slinn, 1980). Mineral dust mobilization mainly depends on vegetation cover, surface soil moisture content, and wind friction speed (Mahowald et al., 2014; Zender et al., 48 49 2003). Once mobilized by wind and lofted into the troposphere, mineral dust can be transported 50 thousands of kilometres from source regions following major air mass movements. In the North 51 Atlantic the trade winds transport dust from North Africa to the Caribbean and Americas (Prospero 52 et al., 2010) in about one week (Ott et al., 1991). Along the transport path of atmospheric aerosols, 53 the size distribution of aerosol tends to decrease with increasing distance from the source regions. 54 Chemical and physical transformation processing of aerosol in clouds, such as photo-reduction and 55 dissolution, can enhance the fraction of trace metals that are released upon deposition into the 56 surface ocean (Duce and Tindale, 1991). Thus, the degree of atmospheric processing can be a critical 57 factor in determining the impact of atmospheric deposition on marine ecosystems (Baker and Croot, 58 2010; Mahowald et al., 2011).

In the Atlantic Ocean, mineral dust deposition is highest in tropical regions located downwind of the 60 major aerosol source regions of the Sahara Desert and Sahel (Jickells et al., 2005; Prospero and 61 Carlson, 1972). Moreover, the Intertropical Convergence Zone (ITCZ) in the Atlantic Ocean (located 62 at  $\sim 5 - 10$  °N in winter and summer, respectively) features intense wet deposition which effectively 63 strip the aerosols from the atmosphere, thereby depositing trace elements to the surface ocean in 64 solution (Kim and Church, 2002; Schlosser et al., 2014). Other important mineral dust source regions 65 include the Namib Desert and Patagonia for the South Atlantic Ocean (Chance et al., 2015; 66 Mahowald et al., 2005; Wagener et al., 2008). In remote marine areas such as the far North and

South Atlantic, removed from major desert dust sources, aerosols are of a mixture of marine origin,
shipping emissions, industrial and agricultural emissions from the continents (Baker et al., 2013;
Chance et al., 2015; Shelley et al., 2017), seasonal emissions of proglacial till (Bullard et al., 2016)
and occasional volcanic ash emissions (Achterberg et al., 2013). In addition, aerosols in these remote
regions may also have a mineral dust component, depending on the meteorological conditions

(Prospero, 1996a).

73 Atmospheric deposition flux determinations have a relatively high uncertainty (Zender et al., 2003) 74 due to large inter-annual and inter-seasonal variabilities. Several approaches are used to calculate 75 atmospheric deposition fluxes from geochemical tracers or proxies (Anderson et al., 2016). 76 Geochemical methods determine atmospheric deposition fluxes from elemental concentrations in 77 aerosols and/or rain, or chemical concentrations or signatures of tracers of atmospheric deposition 78 in seawater (Anderson et al., 2016). A commonly used geochemical method to determine total (wet 79 + dry) deposition fluxes is to measure elemental concentrations in aerosols collected on filters and in 80 rain water, and multiplying the aerosol concentration data with deposition velocities (dry deposition 81 flux) and the rainwater concentration data by a precipitation rate (wet deposition flux) (Baker et al., 82 2003; Prospero, 1996a; Shelley et al., 2018). Atmospheric dry deposition fluxes are subject to a 2-3-83 fold uncertainty due to the use of a single deposition velocity in the flux calculation (Duce and 84 Tindale, 1991) thus not considering the varying deposition velocity of different mineral dust sizes. 85 However, when size segregated sampling approaches are used more than one deposition velocity is 86 applied, thereby decreasing the uncertainty in the calculated deposition flux. The uncertainty may 87 be even larger for wet deposition fluxes because of the uncertainties associated with precipitation 88 rates and scavenging ratios (Shelley et al., 2018). A different approach that tries to reduce the 89 uncertainties associated with the deposition velocity and precipitation rates uses the inventory of 90 <sup>7</sup>Be in the surface mixed layer and the ratio of trace elements to <sup>7</sup>Be in aerosols to calculate 91 atmospheric deposition fluxes on seasonal timescales (Kadko et al., 2015; Kadko and Prospero, 92 2011). Other approaches used to determine atmospheric fluxes of trace elements use particle 93 collection by sediment traps (Jickells et al., 1998; Kohfeld and Harrison, 2001) or analyse mineral 94 dust tracers in surface ocean waters (Dammshäuser et al., 2011). Modern atmospheric models 95 consider aerosol characteristics (e.g. size distribution, particle type) and field observations to 96 estimate atmospheric deposition fluxes (Mahowald et al., 2005; Zender et al., 2003; Zhang et al., 97 2015). Often, modelling approaches simulate atmospheric deposition fluxes more accurately in 98 regions downwind from the main aerosol sources (Huneeus et al., 2011; Wagener et al., 2008) 99 because the aerosol characteristics, due to extensive datasets in these regions, are better 100 constrained in proximity to their source. Modelled atmospheric deposition fluxes often rely on

- 101 satellite-derived climatologies. The latter climatologies use properties (i.e aerosol optical depth)
- which suffer from cloud coverage and are biased towards clear sky conditions (Huneeus et al., 2011).
- In this manuscript we present surface ocean dAl concentration data for the Atlantic Ocean and use 104 these to calculate atmospheric deposition fluxes using the MADCOW model (Measures and Brown, 105 1996). The strength of this approach is that it can be used to fill gaps where there are relatively few 106 direct aerosol observations. This study compares the MADCOW model outputs with field and model-107 derived atmospheric deposition flux estimates from the North to the South Atlantic Ocean. We 108 provide some of the first atmospheric deposition fluxes based on high-resolution surface mixed layer 109 concentrations of dAl for remote regions including the Labrador Sea, South East Atlantic and South 110 Atlantic Ocean (ca. 40°S). Our results are discussed in light of the assumptions and limitations of the 111 MADCOW model, and compared to available atmospheric deposition flux estimates from modelling 112 and geochemical approaches.

#### 113 2. Methods

# 114 **2.1** Regional, sampling, and processing settings

- Seawater samples for dAl were collected during the GEOTRACES section cruises GA01, GA06, GA08, 116 and GA10 (Figure 1). GEOTRACES section GA01 (Sarthou et al., 2018) sailed aboard the research 117 vessel Pourquoi Pas? on 15 May (2014) from Lisbon (Portugal) and arrived on 30 June (2014) in St. 118 John's (Canada). GEOTRACES section GA06 sailed aboard RRS Discovery on 7 February (2011) from 119 Tenerife (Canary Islands, Spain) and returned on 19 March (2011) to Tenerife. GEOTRACES section 120 GA08, on board FS Meteor, sailed from Walvis Bay (Namibia) on 14 November (2015) and returned 121 there on 27 December (2015). GEOTRACES section GA10, on board RRS James Cook, sailed on 24 122 December (2011) from Port Elizabeth (South Africa) and arrived on 27 January (2012) in Montevideo 123 (Uruguay). The four expeditions crossed several biogeochemical provinces (Longhurst, 2010) which 124 are shown in Figure 1 and listed in Table S1 along with their geographical boundaries, ecological and 125 physical properties. Figure S1 shows the station numbers associated with each cruise.
- In total, seawater samples were collected at 108 stations, with 32 stations sampled during GA01, 14
  stations during GA06, 52 stations during GA08 and 18 stations during GA10. Although the different
  teams used slightly different CTD set ups for sampling the seawater (Table S2), all samples were
  collected using trace metal clean CTD rosettes and following the GEOTRACES sampling protocols
  (<u>http://geotraces.org</u>, last access 1 January 2018). All seawater samples for the determination of dAl
  were filtered (Table S2 for filter type and pore size) and collected in 125 mL low density polyethylene
  bottles (LDPE; Nalgene) cleaned using a three-step wash protocol (as per GEOTRACES cookbook;

- http://www.geotraces.org, last access:1 January 2018). After collection, the samples were acidified
- to  $pH \approx 1.8$  with ultra clean hydrochloric acid (UpA, Romil, 0.024M) and double bagged until analysis.
- Analysis of dAl during GA01, GA06, and GA10 used the flow injection analysis (FIA) method
- developed by Resing and Measures (1994) and further modified by Brown and Bruland (2008).
- Dissolved Al during GA08 cruise was analysed using the batch lumogallion method (Hydes and Liss,
- 1976). The analytical figures of merit for the dAl datasets can be found in Table S2.

#### 139 **2.2** Atmospheric deposition flux determinations: The MADCOW model

The MADCOW (Measurement of Aluminium for Dust Calculation in Ocean Waters) model (Measures and Vink, 2000; Measures and Brown, 1996) determines total (dry + wet) atmospheric deposition fluxes to the surface ocean from the concentration of dAl in the surface mixed layer. The primary model assumption is that dAl in the surface waters is in steady state with respect to inputs from soluble Al provided by the dissolution of mineral dust on contact with seawater and rain deposition, and removal via scavenging of Al onto particle surfaces, and subsequent transfer to depth by sinking. The model itself is provided by the following equation:

Eq.1
$$G = \frac{[Al] \times MLD}{\tau \times S \times D}$$

Where G is the total dust flux in g m<sup>-2</sup> yr<sup>-1</sup>, [AI] is the concentration of dAI (mol m<sup>-3</sup>) in the surface mixed layer, MLD is the depth of the mixed layer in meters (m),  $\tau$  is the residence time in years (yr), S 149 150 is the fractional solubility of Al in dust (%), and D is the concentration of Al in dust (8.1 %, mol  $g^{-1}$ ). 151 The limitations of the MADCOW model and extended discussions on the inherent assumptions of the 152 MADCOW model have been acknowledge in previous investigations (e.g. Measures and Vink, 2000; 153 Measures and Brown, 1996). We describe how each parameter is derived for each of the cruises in 154 the following sections and describe the assumptions used by the MADCOW model for the 155 determination of the atmospheric deposition fluxes.

# 156 **2.2.1 Quantification of surface mixed layer depths**

- We used two different MLDs. First, a measured mixed layer depth (MLDms) was obtained for each
- station using a potential density difference criterion  $\Delta\sigma \theta = 0.125$  kg m<sup>-3</sup> (Monterey and Levitus,
- 1997) calculated from the salinity and temperature retrieved from the sensors mounted in the CTDs.
- Second, annual (MLDar) and seasonal (MLDwi, MLDsp, MLDsu and MLDau, for winter, spring,
- summer, and autumn, respectively) MLDs were extracted for each station of the four cruises in a 1 x
- 1 degree bin in latitude and longitude from the Argo mixed layer climatology
- (<u>http://mixedlayer.ucsd.edu/</u>) (Holte et al., 2017).

#### 164 2.2.2 Fractional solubility of Al

Seasonal (April to June and September to November) Al fractional solubilities were obtained from 166 the compilation of Baker et al. (2013), based on aerosol samples collected at high spatial resolution 167 over several years in the Atlantic Ocean. The soluble fraction was based on the results of an 168 ammonium acetate leach at pH 4.7 for 1-2 h, following Sarthou et al. (2003). Baker et al. (2013) 169 divided the Atlantic Ocean into various regions and sub-regions (Figure S2) based on air mass back 170 trajectories and the relative contribution of the air masses over the two seasons in each region. 171 Along with Figure S2 we provide an explanatory note on the relative contribution of each air mass 172 for each region which we then compiled in Table S3. They calculated the fractional Al solubility for 173 each sub region. Baker et al. (2013) provide a good spatial coverage of the Atlantic Ocean, thus 174 allowing us to assign values of Al fractional solubility for different biogeochemical regions (Figure 1) 175 based on field observations. However, the aerosols Al fractional solubility might not be 176 representative over an annual timescale due to, for example, the pulses of Saharan dust and where 177 the mineral dust falls which is related to the position of the ITCZ. Yet, it is still the largest dataset on 178 aerosol Al fractional solubility over the Atlantic Ocean. We produced a single weighted averaged 179 aerosol Al fractional solubility percentage value for each area of interest based on the Al fractional 180 solubility for each air mass to the relative contribution of that air mass to a certain area (Table S3). 181 Furthermore, we assumed that the chosen Al solubility was representative for the whole annual 182 cycle as our atmospheric aerosol deposition flux estimates are given in g  $m^{-2}$  yr<sup>-1</sup>. As the compilation 183 of Baker et al. (2013) does not cover regions north of 50°N in the North Atlantic, we used Al 184 fractional solubilities from (Shelley et al., 2018) derived from aerosol samples collected during the 185 GA01 cruise.

# 186 **2.2.3 Al residence time in the surface mixed layer**

The residence time is defined as the ratio of the dAl inventory in the surface mixed layer to the rate of input or removal. Residence times of dAl in the surface mixed layer were obtained from Han et al. 188 189 (2008) and are based on an extensive surface water dAl observational database (including 22 cruises 190 in the Atlantic Ocean). The latter residence times were derived using a fixed mixed layer depth (50 191 m) and a constant Al fractional solubility (5%) and take into account Al sources to the surface mixed 192 layer from atmospheric deposition, advection, and mixing. The MADCOW model does not account 193 for advection and mixing processes which, for example, occurs in equatorial regions (van Hulten et 194 al., 2013). Thus, the use of modelled residence times which account for advection and mixing 195 sources could partially counterbalance the error associated with our modelled atmospheric 196 deposition fluxes. We acknowledge that the use of residence times derived from using a fixed mixed

layer depth and constant Al solubility may under-over-estimate the real residence time. The latter
affects regions of large seasonal variability as the tropical Atlantic (see sections 3.2.2 and 3.6.2).

## 199 3. Results and Discussion

## 200 **3.1 Mixed layer depth (MLD)**

The surface mixed layer is considered a quasi-homogenous layer based on physical properties (salinity and temperature). The properties display gradients at the bottom of the layer. The bottom depth of the surface mixed layer varies due to atmospheric forcing, with turbulent mixing caused by wind stress (Risien and Chelton, 2008), convection caused by heat exchange (Yu and Weller, 2007), in addition to salinity changes due to evaporation and precipitation at the surface (Schanze et al., 2010). Thus, the thickness of the MLD is an indication of the amount of water that directly interacts with the atmosphere.

Figure 2 shows a box whisker plot with the quantified MLDs (MLD<sub>ms</sub> and MLD<sub>ar</sub>) for each of the four study regions. Table S4 shows the calculated MLDs for each station and Figure S3 (a,b,c,d) shows a comparison plot for each cruise between MLDms, MLDar, MLDmw (mixed layer depth original MADCOW model) and the Argo average MLD during the season when each cruise took place.

In the North Atlantic (GA01) and South Atlantic (GA10) Ocean, large differences were observed between median MLDms and MLDar with a difference of 77 and 20 m, respectively. Smaller 213 214 differences were detected between median MLDms and MLDar in the tropical (GA06) and South East Atlantic (GA08) Ocean with a difference of 7 and 10 m, respectively. Maximum MLDar and MLDms 215 216 were greatest in the North Atlantic (up to 218 m), followed by the South Atlantic (106 m), the South 217 East Atlantic (83 m), and tropical Atlantic (66 m). The large difference, both in maximum MLDar and 218 in the difference between the median MLDms and MLDar, between the North Atlantic Ocean and 219 the other three study areas was due to strong deep mixing taking place in the North Atlantic Ocean 220 during winter (Kara et al., 2003). In the tropical Atlantic Ocean, the MLD is largely controlled by 221 changes in temperature (net heat flux) and salinity (evaporation to precipitation ratios) (Chahine, 222 1992; Webster, 1994). In our study, deeper MLDs relative to shallower MLDs imply that higher 223 quantities of aerosols need to be supplied to a specific region in order to maintain the observed dAI 224 concentrations in the MLD. As input parameter for the MADCOW model we have chosen a single 225 MLD value for each cruise. The latter is the median value between the MLDms and MLDar. We acknowledge that choosing a single MLD value may not be the best approach but it gives us the 226 227 opportunity for intra comparison of atmospheric fluxes within the same cruise.

# 228 **3.2 Distributions of dAl in the surface mixed layer**

Figure 3 shows the average dAl concentration in the surface mixed layer along the four sections.Occasionally, when no sample was collected, the closest underway surface water sample collected

- with a tow-fish was used. A detailed description for the surface mixed layer dAl concentrations of
- the four cruises relative to physical and biological parameters is given in the following sections.
- Overall, a large range in surface dAl concentrations was observed, ranging between < 0.5 nM and
- 784 nM (median 5.1 nM; n=108 stations). The highest dAl concentrations were observed in the
- tropical Atlantic Ocean between 1-15° N and ca. 26° W, and in coastal waters off Portugal,
- Greenland, Argentina, Angola, Democratic Republic of Congo, and Gabon in association with the
- highest atmospheric inputs and continental inputs (rivers, glacial flour, and ice melt). In contrast, low
- dAl concentrations (< 5 nM) due to low atmospheric deposition and/or scavenging of dAl by particles
- were found in the North Atlantic (GA01), in the South East Atlantic (GA08), and in the South Atlantic
- Ocean (GA10).

#### 241 **3.2.1** Dissolved Al in the surface mixed layer of the North Atlantic Ocean (GA01)

Along GEOTRACES GA01 section in the North Atlantic (Figure 1), highest (> 15 nM) dAl surface mixed layer concentrations were found in the North Atlantic Subtropical Gyre region (NAST) off Portugal 243 244 (stations 1, 2 and 4) and are attributed to riverine inputs from the Tagus estuary (Menzel Barraqueta 245 et al., 2018). An additional source of dAl off Portugal involves episodic deposition of mineral dust 246 originating from the Sahara and Sahel regions (Prospero, 1996a), and wet deposition events as 247 observed during the GA01 cruise (Shelley et al., 2017). Enhanced dAl surface mixed layer 248 concentrations (> 5 nM) were also observed in the Atlantic Arctic region (ARCT) off South East and 249 South West Greenland (stations 53 and 61) as a consequence of runoff and ice melt, respectively 250 (Menzel Barraqueta et al., 2018). In addition, enhanced dust inputs delivered from proglacial tills in 251 Greenland occur from June to September (Bullard et al., 2016), coinciding with the time of sample 252 collection off Greenland. Excluding the stations with continental Al inputs, mixed layer dAl 253 concentrations were low, reflecting low contributions of atmospheric deposition (median dAl = 2.9 254 nM), and were not significantly different from east to west due to differences in the intensity of 255 biological removal processes as described by Menzel Barraqueta et al. (2018). A more detailed 256 explanation on the surface distribution for dAl along the GA01 cruise and comparison against 257 previous studies is given in Menzel Barraqueta et al. (2018).

# 258 **3.2.2** Dissolved Al in the surface mixed layer of the tropical Atlantic Ocean (GA06)

The tropical Atlantic has a large coverage of dAl measurements, compiled in Han et al., 2008, related
to the importance of dust delivering micronutrients (e.g. Fe) to the surface ocean. Along the

261 GEOTRACES GA06 section, dAl concentrations were high and ranged from 8 nM in the North Atlantic 262 Tropical Gyre region (NATR) to 67 nM in the Western Tropical Atlantic region (WTRA). In the 263 southern part of the section at~8°S, at the boundary between the South Atlantic Gyre (SAG) and the 264 South Equatorial Current (SEC) in the WTRA (Figure 1), dAl displayed low concentrations (~8.8 nM) 265 with elevated salinity (36.54) (Figure 4). The latter region is known to have low rainfall, high rates of 266 evaporation (Yoo and Carton, 1990) and receives large volumes of mineral dust deposition 267 (Prospero, 1996b). Vink and Measures (2001) observed similarly low levels of dAl (8 nM) but 268 somewhat further south at 15°S. Similarly, on the westward transect at ca. 12.5°N, low dAl 269 concentrations were observed (down to 8 nM), associated with enhanced removal of dAl by biogenic 270 particles. Enhanced primary production as a consequence of upwelling of nutrient-rich deep water 271 resulted in a high abundance of biogenic particles; Measures et al. (2015) reported similar conditions 272 for the region. Also, the low levels observed could be associated to the westward transport of 273 depleted Al waters arising from the African upwelling region (Gelado-Caballero et al., 1996). High dAl 274 concentrations (15-28 nM) were found north of 3°S and were related to enhanced deposition of 275 mineral dust as the sampling stations were located along the flow path of the trade winds that carry 276 mineral dust from the Sahara and Sahel regions (Mahowald et al., 2005; Prospero et al., 2002). 277 Maximum concentrations of dAI (up to 68 nM) were observed at ca. 3°N, 26° W (station 14) and 278 coincided with reduced salinity (down to 35) (Figure 4), suggesting freshwater inputs from rainfall 279 occurring in the ITCZ. The ITCZ was positioned between ca. 3°S and 3°N with the core situated at 1°N 280 during the cruise period (Schlosser et al., 2014). Precipitation in the ITCZ effectively scavenges dust 281 from the atmosphere and supplies Al to surface waters in the form of wet deposition (Schlosser et 282 al., 2014). Similarly, high dAl concentrations (or total Al, e.g. up to 74 nM) (Van Der Loeff et al., 1997) 283 for the region have been reported (Barrett et al., 2015; Bowie et al., 2002; Dammshäuser et al., 284 2011; Helmers and Rutgers van der Loeff, 1993; Measures, 1995; Measures et al., 2015; Measures et 285 al., 2008; Measures and Vink, 2000; Pohl et al., 2011; Schlosser et al., 2014). The latter studies either 286 relating the high Al values to wet deposition and/or large dry deposition pulses.

287 Figure 5 shows average, maximum and minimum Al concentration data (filtered and unfiltered) 288 collected over several cruises from approximately 25°S to 25°N. The single data sets used for the figure are shown in figure S4. From figure 5 and S4 we can observe that Al values between 8° S and 289 290 17° N are highly variable with a large range between maximum and minimum values. The latter 291 variability is a consequence of seasonal migration of the ITCZ and the seasonal nature of mineral 292 dust plumes which vary in intensity on a daily basis (Patey et al., 2015). The latter factors, in addition 293 to equatorial upwelling (Vink and Measures, 2001), will impact particle fluxes (biogenic and non-294 biogenic) (Chester, 1982; Kuss et al., 2010) which, in turn, impact the residence time of Al in the

tropical Atlantic and our ability to accurate calculate atmospheric fluxes (see section 3.6.2). The

296 influence of sporadic inputs of AI (either wet or dry) is critical as it has been demonstrated that dAI

concentrations in surface waters can change substantially within weeks after sporadic inputs (de

Jong et al., 2007; Schlosser et al., 2014). Deciphering the seasonal variability of dAl in the tropical

- Atlantic seems a difficult task. The latter, if done by only using Al surface concentrations, would
- struggle to yield precise outcomes.

# 301 **3.2.3** Dissolved Al in the surface mixed layer of the South East Atlantic Ocean (GA08)

Along GEOTRACES section GA08 in the South East Atlantic, dAl concentrations (Figure 3) ranged from 303 1.2 (station 45) to 784 nM (station 15) (median 9 nM; n=44). The lowest dAl concentrations were 304 found south of 20°S at stations along the prime meridian and at ca. 30°S and 1° to 10°E as a 305 consequence of low aerosol deposition to the South Atlantic Gyre (SAG) region. Similar low 306 concentrations of surface dAl associated with low productivity waters in the South Atlantic Gyre 307 have been reported (Measures, 1995). Low dAl concentrations (down to 1.3 nM) were observed 308 during GA08 in the Benguela Coastal Current (BENG) region off Namibia coinciding with high Chl a 309 concentrations (Figure S5) and thus enhanced dAl scavenging onto biogenic particles. The enhanced 310 scavenging of dAI by particles is supported by elevated concentrations of total dissolvable AI 311 (unfiltered) during the AMT-6 cruise (Bowie et al., 2002). Bowie et al. (2002) argued that the 312 enhanced total dissolvable AI was a consequence of AI-rich upwelled waters. Significant correlations between increased number of biogenic particles, and low dAl and high pAl concentrations have 313 314 recently been reported (Menzel Barraqueta et al., 2018; Barret et al., 2018), indicating the control of 315 biogenic particles on Al cycling in surface waters. The highest dAl concentrations (up to 784 nM) 316 were observed along the SW African coast north of 6°S (stations 14 to 23) in the Guinean Current 317 Coastal region (GUIN) and were associated with freshwater inputs from the Congo River. Similar observations were made by van Bennekom and Jager (1978) and enhanced levels of dAl (up to 50 318 319 nM) associated with inputs from the Congo river have been reported as far as 1200 km (6°S 0°E) 320 from the river mouth (Van Der Loeff et al., 1997). Between the Congo River mouth (6°S) and the 321 Angola-Benguela frontal zone (ca. 11°S) (Figure 1) elevated dAl was observed (20 – 40 nM) (Figure 3). 322 Possible sources for the elevated dAl concentrations are urban emissions and atmospheric mineral 323 dust from West Africa. Prospero (1996a) argued that when the ITCZ migrates south of the equator, 324 additional southward transfer to the Gulf of Guinea of atmospheric dust from the Sahara and the 325 Sahel regions can occur during austral summer. In addition, the low Chl a concentrations in the GUIN 326 region would facilitate dAl accumulation in the surface mixed layer due to a lack of adsorption onto 327 biogenic particles.

#### 328 **3.2.4** Dissolved Al in the surface mixed layer of the South Atlantic Ocean (GA10)

Along GEOTRACES section GA10 in the South Atlantic Ocean, dAl concentrations were generally low 330 and ranged between 0.3 nM in the SAC and 15.8 nM on the Argentine shelf (n=17, median = 1.6 nM) 331 (Figure 3). Highest concentrations (up to 16 nM) were observed on the South West Atlantic Shelf 332 region (FKLD). The FKLD is thought to be influenced by dAl inputs from the River de la Plata, shelf 333 sources, and possibly atmospheric deposition from Patagonian sources (Chance et al., 2015; Jickells 334 et al., 2005; Mahowald et al., 2005; Wagener et al., 2008). However, the correlation between 335 surface salinity and dAl for the FKLD is weak (n=4, R<sup>2</sup>=0.13) and suggests a minor influence on Al concentrations from freshwater sources. Thus, it is unlikely that the River de la Plata was a major 336 337 source of dAI at the time of the cruise. An additional input of dAI could be associated from the 338 advection Al rich waters of the southward flowing Brazil Current (Vink and Measures, 2001). 339 However, this feature has been observed somewhat further north at 34°S and the influence of the 340 Brazil Current at our latitude may been counterbalanced by the northward flowing Malvinas Current 341 (Figure 1). Similar dAl concentrations in the vicinity of the South American continent were observed 342 during expedition ANT IX/1 (Van Der Loeff et al., 1997) and AMT-3 (Bowie et al., 2002) and 343 attributed to recent dust deposition. In the South Subtropical Convergence region (SSTC), the dAl 344 concentrations were low (<5 nM), reflecting low Al inputs from atmospheric deposition (Mahowald 345 et al., 2005) and/or removal via particle scavenging in SSTC along ca. 40° S. Our observations agree 346 with measurements of low concentration of dAl in the high latitude South Atlantic (Middag et al., 347 2011).

# 348 **3.3 Fractional solubility of Al (Al**sol%):

The fractional solubility of trace metals from aerosols is controlled by: 1) chemical processing during atmospheric transport which is influenced by the relative humidity of the particles (Keene et al., 350 351 2002), the balance of acid species (enhanced by anthropogenic sources e.g. fossil fuel combustion; (Ito, 2015; Sholkovitz et al., 2012) and the phase partitioning of NH<sub>3</sub> (Hennigan et al., 2015); and 2) 352 353 composition and type of particle; aerosols from different sources have different mineralogies and 354 size distributions which influence the solubility of metals (Baker and Jickells, 2017). Reported aerosol 355 Al fractional solubilities in aerosols span a large range from 0.5 to 100 % (Baker et al., 2006; Buck et 356 al., 2010; Measures et al., 2010; Prospero et al., 1987; Shelley et al., 2017). Model approach by Han 357 et al. (2012) resulted in a global Al fractional solubility of 4.1 (3.8 and 7 for the North and South 358 Atlantic respectively). However, the latter model uses Al fractional solubility data of 3 cruises due to 359 the lack of available field data at that time. One of the reasons for this large range is likely the lack of 360 standardisation in aerosol leaching methods (Aguilar-Islas et al., 2010). An up to twentyfold (1 to 20

361 g m<sup>-2</sup> yr<sup>-1</sup>) difference between atmospheric deposition fluxes estimated from Al fractional solubility

values following two different aerosol leaching methods has been reported by Anderson et al.

(2016). However, based on aerosol Fe solubility experiments, the largest differences resulted from

aerosols from different sources (Aguilar-Islas et al., 2010; Fishwick et al., 2014).

In an early study which used dAl concentrations in the surface mixed layer to derive atmospheric deposition fluxes, Measures and Brown (1996) choose an Al fractional solubility range of 1.5 to 5 % for the tropical Atlantic, with the upper limit derived from aerosols and rain data by Prospero et al. (1987). Measures and Brown (1996) discussed that the chosen upper boundary for Al fractional solubility may have been too low, as earlier studies (Maring and Duce, 1987) reported an Al fractional solubility of 9 %, attributed to cloud processing during long-range atmospheric transport to the mid-Pacific Ocean, and thus not directly applicable to tropical Atlantic aerosols.

As our study region spans the entire Atlantic Ocean, we expect a large range in Al fractional solubility 373 due to multiple aerosol sources (Baker and Jickells, 2017; Baker et al., 2006). Table S3 shows the Al 374 fractional solubility for each region and sub-region (Figure S2) and the relative abundance of air 375 mass back trajectories. The Al fractional solubility partially reflects the different aerosol sources. The 376 lowest AI fractional solubility values (5-7.7%) were estimated for the NTRA and WTRA in the tropical 377 Atlantic (GA06) and for the GUIN and BENG region in the Southeast Atlantic (GA08) due to the 378 dominance of mineral dust. The highest Al fractional solubility (11.6-21%) was calculated at high 379 latitudes (GA01 and GA10) and in the SATL (GA08) which was likely due to the greater distances from 380 aerosol source regions and/or higher degree of atmospheric processing that these aerosols have 381 undergone during transport (e.g. Shelley et al., 2017). Overall, the Al fractional solubility used in this 382 study ranged from 5 to 20 %. The higher the Al fractional solubility value, the less aerosol material is required to maintain the observed dAl surface mixed layer concentrations. 383

We acknowledge the lack of use of Al solubility values in rain, which has been estimated to account for 89% of total deposition to the ocean (Zender et al., 2003). However, the amount of published data of Al fractional solubility in rain is still minor (Heimburger et al., 2013; Losno et al., 1993). The option to link the few values to global precipitation models (Liu et al., 2012) seems, at this stage, a more qualitatively than quantitatively approach in comparison to the larger amount of Al fractional solubility data available for dry aerosols.

# 390 3.4 Residence time of dAl

The residence time of dAl in the surface mixed layer is a balance between atmospheric, riverine and
 sedimentary inputs, and removal processes which are dominated by scavenging of dAl by biogenic

particles (Orians and Bruland, 1986). Short residence times are associated with regions of enhanced 394 mineral dust deposition (Dammshäuser et al., 2011; Schüßler et al., 2005) such as the tropical North 395 Atlantic and regions of enhanced biological activity (primary productivity) such as the North Atlantic 396 (north of 40°N). In contrast, long Al residence times are associated with regions of low mineral dust 397 deposition and low biological activity (Dammshäuser et al., 2011; Jickells, 1999) such as the South 398 Atlantic subtropical gyre. The residence time of dAl in the upper ocean has been estimated to range 399 from ~0.2 to more than 17 yr (Dammshäuser et al., 2011; Jickells, 1999; Jickells et al., 1994; Orians 400 and Bruland, 1986) and up to 73 yr in modelling studies (Han et al., 2008). In this study, we used 401 short Al residence times in the surface mixed layer for the North Atlantic (GA01), tropical Atlantic 402 (GA06), along the upwelling regions in the Southeast Atlantic (GA08), and for the South Atlantic 403 along the South Subtropical Convergence region (SSTC) (GA10). Longer Al residence times were used 404 for the South East Atlantic Ocean, more precisely along the prime meridian as a consequence of low 405 removal rates removal due to low primary productivity and low atmospheric deposition. Overall, the 406 Al residence times used in this study (Table 1), and derived for each biogeochemical province from 407 the estimates provided in Han et al. (2008), ranged between 0.75 and 3 years.

# 3.5 Application of the MADCOW model to derive total atmospheric deposition fluxes in the study area

#### 410 **3.5.1 MADCOW** input parameter assumptions

The original MADCOW model (Measures and Brown, 1996) assumed a uniform residence time of 5 yr 412 for dAl in the surface mixed layer, an invariant content of Al in dust of 8%, a fixed MLD of 30 m, and 413 an aerosol Al fractional solubility between 1.5 and 5%. The total dust deposition was calculated by 414 multiplying the concentration of dAl in the mixed layer by 0.133 and 0.04 (factors for unit conversion) for a solubility of 1.5 and 5 %, respectively. Our study region spans the whole Atlantic 415 416 Ocean, with a range in MLDs, fractional Al solubilities, and residence times of Al in the surface mixed layer. The different input parameters used for the dust flux calculations for each of the four study 417 418 regions can be found in Table 1 and relate to the factors described in Sections 2 and 3. Mixed layer 419 depths were deeper for GA01 and GA10, and shallower for GA06 and GA08. The residence time of 420 dAl in the surface mixed layer along the four transects was quite uniform and ranged from 0.75 to 3 421 years. The largest variability was noted for the aerosol AI fractional solubility due to inter-annual and 422 inter-seasonal variability in the contribution of different aerosol sources, incomplete spatial 423 coverage, wet to dry deposition balance, etc. As a consequence, for each cruise we have chosen a 424 lower and upper limit of Al fractional solubility. As such we obtained an upper and lower range in 425 atmospheric deposition fluxes for each cruise. The total range on aerosol Al fractional solubility used

in this study was 5 to 20%, which is up to 3x and 4x higher than the lower and upper limit used by
Measures and Brown (1996). Higher Al fractional solubilities were used for GA01 and GA10 and
lower values for GA08 and GA06.

# 429 **3.6 Atmospheric deposition fluxes**

Figure 7 shows the average atmospheric deposition flux (wet + dry) for each station. In addition, Figure 8 (a, b, c and d) shows a comparison of calculated average total atmospheric deposition fluxes 431 432 (this study) and total atmospheric deposition fluxes for each station extracted from the models of 433 Mahowald et al. (2005) and Zender et al. (2003). In Table 2, we present our total atmospheric 434 deposition fluxes values for each biogeochemical province compared to the model estimates from 435 Mahowald et al. (2005), Zender et al. (2003), and Duce et al. (1991). Overall, we found good 436 agreement along GA01, GA08, and GA10. Weaker agreement was found in the tropical Atlantic along 437 GA06. In the following sections we describe our atmospheric deposition fluxes in more detail.

#### 438 **3.6.1** Atmospheric deposition fluxes to the North Atlantic and Labrador Sea (GA01)

In the North Atlantic, along the GA01 section, our calculated atmospheric deposition fluxes ranged 440 from  $1.75 \pm 0.71$  g m<sup>-2</sup> yr<sup>-1</sup> (St. 2) near the Iberian Peninsula in the NAST region to  $0.12 \pm 0.05$  g m<sup>-2</sup> yr<sup>-1</sup> 441 <sup>1</sup> (St. 78) above the Newfoundland margin in the ARCT region (Figure 7 and 8a). The average 442 atmospheric deposition flux was  $0.49 \pm 0.46$  g m<sup>-2</sup> y<sup>-1</sup>. The highest atmospheric deposition fluxes 443 were calculated in the vicinity of land masses (i.e. Iberian Peninsula and Greenland) and could be 444 partly due to overestimations by the MADCOW model. An overestimation would arise if there were 445 additional Al sources, rather than Al being input solely from atmospheric deposition. The coastal 446 stations (1, 2 and 4; Figure S1) near the Iberian Peninsula received additional Al inputs from the 447 Tagus estuary (Menzel Barraqueta et al., 2018), while the coastal stations (53 and 63; Figure S1) near 448 Greenland are influenced by additional Al inputs from glacial run-off and ice melt (Menzel 449 Barraqueta et al., 2018). With these stations removed, the average atmospheric deposition flux 450 along GA01 decreased to  $0.28 \pm 0.12$  g m<sup>-2</sup> yr<sup>-1</sup> (n=24). Although the atmospheric deposition fluxes near the Iberian Peninsula were high relative to the rest of the transect, they are comparable to 451 452 modelled atmospheric aerosol deposition fluxes (Mahowald et al., 2005; Zender et al., 2003) (Table 2 and figure 7). Measures et al. (2015) reported an atmospheric deposition flux of ca. 0.91 g m<sup>-2</sup> yr<sup>-1</sup> 453 454 (station USGT10-02) close to the Iberian Peninsula along GEOTRACES section GA03 (Figure 7b), 455 which is somewhat lower than our average deposition flux for stations 1, 2 and 4 (1.63  $\pm$  0.08 g m<sup>-2</sup> yr<sup>1</sup>). Our calculated fluxes derived from dAl at coastal stations near the Iberian Peninsula may be on 456 457 the high end of the scale due to additional Al inputs from the Tagus estuary which did not influence

the nearby stations sampled by Measures et al. (2015). In the NAST we derived an average dust flux 459 for stations 21, 23 and 25 of  $0.42 \pm 0.17$  g m<sup>-2</sup> yr<sup>-1</sup> which is in good agreement with calculated 460 atmospheric deposition fluxes for the same region during the 2003 CLIVAR A16N cruise (Measures et 461 al., 2015) using aerosol Al concentrations from Buck et al. (2010) (Figure 7b). Shelley et al. (2017) 462 reported atmospheric deposition fluxes for a suite of elements for the GA01 cruise using two 463 different approaches; 1) aerosol and precipitation concentration data (which they termed "traditional approach") and 2) <sup>7</sup>Be in aerosols and the surface mixed layer. They divided the GA01 464 section into two areas: (i) Area 1 = west of 30° W; (ii) Area 2 = east of 30° W (Figure 1). For Area 1 465 they derived atmospheric deposition fluxes of 0.03 and 0.1 g m<sup>-2</sup> yr<sup>-1</sup> using the traditional and <sup>7</sup>Be 466 approaches, respectively. For Area 2, the atmospheric deposition fluxes were 0.19 and 0.14 g m<sup>-2</sup> yr<sup>-1</sup> 467 468 for the traditional and <sup>7</sup>Be approaches, respectively. With the MADCOW model we calculated average atmospheric deposition fluxes of 0.27  $\pm$  0.16 and 0.3  $\pm$  0.12 g m<sup>-2</sup> yr<sup>-1</sup> for Area 1 and 2, 469 470 respectively. The average modelled flux estimates for the same areas from Mahowald et al., 2005 471 were 0.36  $\pm$  0.03 and 0.76  $\pm$  0.32 g m<sup>-2</sup> yr<sup>-1</sup>. It is encouraging that the different flux results are of a 472 similar order of magnitude, although the dust flux estimates derived from Mahowald et al. (2005) 473 were always highest. It is possible that the dust deposition model (Mahowald et al., 2005) may have 474 overestimated atmospheric deposition in the North Atlantic (north of 50°N) due to the limited 475 number of field observations available at the time. Interestingly, a higher atmospheric deposition 476 flux was calculated with the MADCOW model than with either the aerosol approach or the <sup>7</sup>Be 477 method. This could be due to the different timescales over which each approach integrates. The 478 MADCOW model approach integrates the total atmospheric deposition flux over a period of ca. 1 yr 479 prior (based on our choice of dAl surface mixed layer residence time) to the GA01 cruise, while the 480 <sup>7</sup>Be approach integrates over a period of 3 months, and the aerosol approach provides a snapshot 481 over a period of days. The elevated atmospheric deposition flux for the traditional compared to the 482 <sup>7</sup>Be approach in area 2 was attributed to <sup>7</sup>Be scavenging onto biogenic particles near the Iberian 483 Peninsula (Shelley et al., 2017).

## 484 **3.6.2** Atmospheric deposition fluxes to the Tropical Atlantic (GA06)

Overall, the atmospheric deposition fluxes calculated for the tropical Atlantic using MADCOW are
generally of the same order of magnitude compared to modelled atmospheric deposition fluxes
(Mahowald et al., 2005; Zender et al., 2003) (Table 2 and Figure 6, 7b). Along GA06, the calculated
atmospheric deposition fluxes ranged from 1.19 ± 0.45 g m<sup>-2</sup> yr<sup>-1</sup> at station 8 to 9.96 ± 3.72 g m<sup>-2</sup> yr<sup>-1</sup>
at station 14 (Figure S1 and 7, 8b). The average calculated atmospheric deposition flux along the
cruise was 3.46 ± 2.18 g m<sup>-2</sup> yr<sup>-1</sup>. In contrast, in the WTRA, our average atmospheric deposition flux of

$3.82 \pm 2.72 \text{ g m}^{-2} \text{ yr}^{-1}$  agrees very well with modelled atmospheric deposition fluxes (4.4 ± 3.2 and 492  $7.15 \pm 4.35 \text{ g m}^{-2} \text{ yr}^{-1}$  for 'Mahowald' and 'Zender'). However, in the NATR, large discrepancies are 493 observed between our and modelled atmospheric deposition fluxes (Table 2, figure 8b). The largest 494 differences are found along the East-West transect and north of 8°N. Along the East-West transect 495 (Station 7 to 9), at ca. 12°N, and north of 8°N, our atmospheric deposition fluxes ranged between 496  $1.19 \pm 0.44$  and  $3.54 \pm 1.32 \text{ g m}^{-2} \text{ yr}^{-1}$  while modelled dust fluxes were above 8 g m $^{-2} \text{ yr}^{-1}$  and reach up 497 to 12 g m $^{-2} \text{ yr}^{-1}$  (Figure 8b).

The discrepancy could result from differences in the solubility of aerosols in the east and west of the 499 basin. A combination of more soluble aerosols transported from mineral dust sources regions in 500 North Africa via the trade winds to the western tropical Atlantic and from American sources may 501 result in higher levels of dAl in the western tropical Atlantic than in the eastern tropical Atlantic 502 (Measures et al., 2015; Sedwick et al., 2007). In addition, the Mahowald model uses extensive field 503 data and monthly satellite retrievals over many years. Thus, the latter model may better capture the 504 high degree of inter-annual and inter-seasonal variability in dust. Whilst there is more dust being 505 deposited in the eastern tropical Atlantic (e.g 18 gr m<sup>-2</sup> yr, annual average, Powell et al., 2015) than in the western tropical Atlantic (e.g 1.71 gr m<sup>-2</sup> yr in the Sargasso Sea, Jickells et al., 1994), MADCOW 506 507 calculates higher atmospheric deposition fluxes in the western tropical Atlantic than in the eastern 508 tropical Atlantic (this study and Measures et al., 2008; 2015).

The NATR region is influenced by strong seasonal variations (e.g. Intensity of rainfalls, episodic dust 510 events) which influence primary productivity (i.e. number of particles). Therefore, the latter aspects 511 have a strong influence in seasonal variability in Al concentrations (Pohl et al., 2011) as well as 512 seasonal variability of particle fluxes (i.e. potential reducers of dAl concentrations in surface waters). 513 This implies an added difficulty in assessing a residence time for Al. The residence time in the NATR 514 could be lower, in the range of weeks, as postulated in previous works (Croot et al., 2004). Assuming 515 a residence time of 0.3 yr (Original residence time set up to 1.25 yr) for the NATR would yield 516 atmospheric deposition fluxes ranging between  $4.96 \pm 1.86$  and  $19.37 \pm 7.25$  g m<sup>-2</sup> yr<sup>-1</sup> (Average 517  $12.32 \pm 4.66$  g m<sup>-2</sup> yr<sup>-1</sup>) which, within uncertainty, are equal to those provided by modelling 518 approaches (see above, Mahowald, 2005; Duce et al, 1991, and Zender et al., 2003).

These results do not match the observations (from field data and satellite retrievals) and suggests that atmospheric deposition fluxes calculated with the MADCOW model are less reliable and likely underestimated in the tropical North Atlantic Ocean if seasonal variations in the residence time of Al are not accounted for.

### 523 **3.6.3** Atmospheric deposition fluxes to the South East Atlantic (GA08)

The MADCOW model overestimates atmospheric aerosol deposition fluxes to the GUIN region north 525 of 6°S with fluxes ranging between 38 to 163 g m<sup>-2</sup> yr<sup>-1</sup>. We have omitted stations 14 to 21 from the 526 average atmospheric deposition flux, as they are strongly influenced by additional dAl inputs from the Congo River (Figure 3 and S1). Following this removal, atmospheric deposition fluxes along GA08 527 528 (Figure 7, 8c) ranged from  $0.04 \pm 0.01$  g m<sup>-2</sup> yr<sup>-1</sup> in the South Atlantic Gyre region (SATL) to 7.08 ± 529 1.44 g m<sup>-2</sup> yr<sup>-1</sup> in the Guinea Current Coastal region (GUIN) at ca. 12° S (Table 2). The average calculated atmospheric deposition flux to the South East Atlantic was  $1.33 \pm 1.85$  g m<sup>-2</sup> yr<sup>-1</sup>. The 530 531 MADCOW model may underestimate atmospheric deposition fluxes in the BENG region, which is 532 influenced by eastern boundary upwelling processes, supplying nutrients from deep to surface 533 waters and resulting in enhanced primary productivity with high levels of Chl a (Figure S5). The BENG 534 region receives mineral dust inputs from the Namib Desert (Prospero et al., 2002), but the large 535 abundance of biogenic particles likely results in low surface dAl concentrations (< 7 nM) (Figure 3) 536 due to enhance scavenging. Mahowald et al. (2005) calculated an average atmospheric deposition 537 flux for the BENG region of  $5.2 \pm 4.16$  g m<sup>-2</sup> yr<sup>-1</sup> with deposition values of up to 11.96 g m<sup>-2</sup> yr<sup>-1</sup> (station 48). However, our atmospheric deposition fluxes and Zender's ones for this region are 538 539 rather low with an average of  $0.36 \pm 0.18$  g m<sup>-2</sup> yr<sup>-1</sup> and  $0.1 \pm 0.09$  g m<sup>-2</sup> yr<sup>-1</sup>, respectively. In contrast, our MADCOW derived atmospheric deposition fluxes in the SATL region ( $0.17 \pm 0.18 \text{ g m}^{-2} \text{ yr}^{-1}$ ) are in 540 close agreement with model estimates ( $0.22 \pm 0.09 \text{ g m}^{-2} \text{ yr}^{-1}$ ). 541

#### 542 **3.6.4** Atmospheric deposition fluxes to the South Atlantic Ocean (GA10)

Along GA10, the calculated atmospheric deposition fluxes ranged from  $0.15 \pm 0.1$  g m<sup>-2</sup> yr<sup>-1</sup> in the 544 South Subtropical Convergence (SSTC) region to  $1.23 \pm 0.67$  g m<sup>-2</sup> yr<sup>-1</sup> in the South West Atlantic Shelf 545 (FKLD) region off Argentina. The average atmospheric deposition flux calculated along the 40°S section was 0.21 g m<sup>-2</sup> yr<sup>-1</sup>. The highest atmospheric deposition flux was estimated in the FKLD (1.23 546 547 ± 0.67), downwind from South America with Patagonian dust reported as the main source of mineral 548 dust to the South Atlantic Ocean (Johnson et al., 2010; Wagener et al., 2008). Our atmospheric 549 deposition fluxes in the FKLD agree with model estimates for the same region from Mahowald et al. 550 (2005) (Table 2, figure 7, 8d) and Wagener et al. (2008). In contrast, the 'Zender' model atmospheric 551 deposition flux within the FKLD  $(0.21 \pm 0.03)$  is significantly lower than the deposition flux from our 552 study, Mahowald et al. (2005), and Wagener et al. (2008). Some discrepancies exist between the 553 region of maximum atmospheric deposition. Indeed, Wagener et al. (2008) acknowledged this issue 554 stating that the largest uncertainty in their estimates occurred downwind from South America as their field data were collected on cruises in the South Pacific and South Indian Oceans. The lowest 555

atmospheric aerosol deposition was found along the SSTC region ( $0.15 \pm 0.1$  g m<sup>-2</sup> yr<sup>-1</sup>) and was 557 somewhat lower than model results but identical within the calculated uncertainties (Table 2). In the EAFR, we calculated an average atmospheric deposition flux of  $0.21 \pm 0.22$  g m<sup>-2</sup> yr<sup>-1</sup>. The latter flux 558 559 agrees very well with total atmospheric deposition fluxes for the same region calculated from 560 aerosol samples collected on the Falkland Islands (0.18 g m<sup>-2</sup> yr<sup>-1</sup>) (Chance et al., 2015) and with 561 model results (Mahowald et al., 2005; Zender et al., 2003). It is possible that the South Atlantic Ocean also receives additional mineral dust inputs from South African sources although Patagonian 562 dust is considered the major source of aerosols to this region (Gaiero et al., 2003; Wagener et al., 563 564 2008).

# 566 4 Conclusions

Dissolved Al concentrations in the surface ocean are a balance between input and removal processes. In this context, we can use the concentration of dAl in the mixed layer along with the 568 569 residence time of dAl in the mixed layer to calculate atmospheric deposition fluxes. Overall, in 570 oceanic regions beyond the shelf break, we found good agreement between our calculated 571 MADCOW atmospheric deposition fluxes and modelled atmospheric deposition fluxes (Mahowald et 572 al., 2005; Zender et al., 2003). The agreement between our MADCOW deposition flux calculations 573 and model results was poor in regions with additional Al inputs (e.g. river run off, ice melt, and 574 benthic) and strong AI removal by biogenic particles in upwelling regions. Our atmospheric 575 deposition fluxes were lower than model fluxes in areas of the Atlantic Ocean regions removed from 576 the main aerosol sources regions. This observation suggests that these regions receive less 577 atmospheric inputs than the models indicate or that MADCOW underestimates atmospheric inputs 578 to these regions. As such, this work provides new constraints for models of atmospheric deposition 579 for the largely under-sampled regions of the Atlantic Ocean (e.g. Labrador Sea, South Atlantic and 580 Southeast Atlantic Ocean). Specifically, Mahowald et al. (2008) note that there are few aerosol 581 measurements reported for the region between 30° and 60° S, due to a lack of island sites for 582 deployment of aerosol samplers and the remoteness of the region presenting logistical challenges 583 for research cruises. In the GEOTRACES programme, a concerted effort has been made to cover 584 these under-sampled regions of the ocean. Dissolved Al is a key trace element of the GEOTRACES 585 programme and as such it is measured on all the GEOTRACES cruises which implies a great chance to 586 use the MADCOW model. We acknowledge that there are regions of the Atlantic for which the 587 application of the MADCOW model has limitations (i.e. tropical Atlantic Ocean). In these regions, 588 more work is needed to constrain parameters such as aerosol AI fractional solubility and the

- residence time of Al within the surface mixed layer. Realistic residence time values would
- significantly improve our atmospheric deposition flux results. We suggest that forthcoming
- expeditions should use more than one technique to calculate atmospheric deposition fluxes as
- different methods provide us with complimentary information. The new Intermediate Data Product
- 2017 (Schlitzer et al., 2018) from the GEOTRACES Programme along with upcoming dAI datasets will
- give us a better coverage of under-sampled regions which combined with historical data will help us
- to refine atmospheric deposition fluxes (i.e. capturing seasonal and inter-annual variability).

# 596 Acknowledgments

- We are greatly thankful to the captains and crews of expeditions GA01, GA06, GA08 and GA10 for
- their support during the cruises. Special thanks goes to the trace metal colleagues on board GA01,
- GA06, GA08, and GA10. We thank all three anonymous reviewers and Gideon Henderson and
- Laurent Boop for editorial work. We also want to thank Charlie Zender and Nathalie Mahowald for
- providing guidance on their model outputs. This work was funded by a PhD fellowship to Jan-Lukas
- Menzel Barraqueta from the Department of Scientific Politics of the Basque Government, with
- further financial support from GEOMAR Helmholtz Center for Ocean Research Kiel. GEOTRACES
- GA01 was led by France, GA06 and GA08 by the UK, and GA10 by Germany.
