# Peer review of "Atmospheric deposition fluxes over the Atlantic Ocean: A"

_Biogeosciences, 2018_

## Referee Comment (RC4)

**Review: bg-2018-209**

**Overview:**

This manuscript presents results of the application of the MADCOW model for aerosol deposition to recent GEOTRACES data from the Atlantic. The authors expand on the original MADCOW model by varying previously fixed parameters through a combination of comparison with field data for fractional solubility and model date for residence times. While it is an interesting topic, much of the discussion reads like a summary of the earlier works and the manuscript would be better focused on providing new insights into the GEOTRACES datasets through examining how well the assumptions in the MADCOW model are adhered to. There are some question marks regarding the GA08 AI data set also as it the dissolved AI values appear to be overestimated possibly due to the lack of correction for CDOM fluorescence due to the methodology that was used during that expedition. Overall this paper does a good job in adding value to existing GEOTRACES datasets and could make a very useful contribution to this field if it is revised along the lines outlined below.

**General Comments:**

**Atmospheric fluxes – wet and dry**

While the paper does a reasonable job of explaining how the fluxes were calculated it does not get into a detailed comparison with atmospheric based fluxes for which there is also data from GEOTRACES and other programs. One aspect of the current work where the atmospheric data would help decipher things is in assigning how much of the surface Al comes from aerosol flux (dry deposition) and how much from wet deposition. In this regard making the link to the precipitation fluxes for each region (Liu et al., 2012) would be beneficial in examining if this is what determine the high inferred model solubility of the aerosols or not. As the assumption of the MADCOW model is that dry deposition is the only process occurring and that in areas where wet deposition is important a higher fractional solubility is assumed. There are data for aluminium solubility in marine rain (Heimburger et al., 2013; Losno et al., 1993), the Losno et al. (1993) paper includes several samples from the Atlantic. See also for example the impact of the Saharan air layer and the ITCZ on the relative humidity in the atmosphere (Braun, 2010). Addition of this type of analysis would greatly increase the impact of this work.

**Seasonality and residence time:**

A critical weakness of simple box models like the MADCOW model is that areas with strong seasonality of inputs/outputs are inadequately described when using a single concentration term to fix the inventory. Previous work has indicated that the seasonal cycle (or interannual variability) off the west African coast is on the order of 60 nM for dissolved Al (Pohl et al., 2011) and presumably the residence time is then shorter than 2-5 years first postulated by Helmers and van der loeff (1993). Indeed comparison with Fe suggests that the residence time could be much less than a year or so (Croot et al., 2004; Dammshäuser, 2012; Dammshäuser and Croot, 2012) in these high dust impacted regions.

At present there is little discussion regarding the assumptions inherent in a steady state model such as MADCOW, the focus in the paper is on the inventory size as determined by mixed layer depth and concentration and not on whether the fluxes are in balance over the time scales being investigated. In this regard there are a number of studies that have looked at the seasonality of particle fluxes of Al in the North Atlantic (Chester, 1982; Hwang et al., 2010; Hwang et al., 2009; Jickells, 1999; Jickells et al., 1984; Kuss and Kremling, 1999a; Kuss et al., 2010). With regard to the seasonality in the Benguela region, there has been recent work looking at the fluxes from the Namib (Dansie et al., 2018; Dansie et al., 2017a; Dansie et al., 2017b) and their predominance during austral winter that is of relevance here to the question of inputs and residence times.

The challenge that arises then is how to reconcile a snap shot residence time provided by a single concentration measurement within a very active seasonal cycle. For example most sampling is in summer which while likely to be the maximum sink for dissolved AI due to enhanced biological productivity and scavenging, but also could be a minimum in atmospheric deposition leading to a residence time of weeks. Contrastingly winter measurements may have higher deposition rates and minimal scavenging resulting in longer apparent residence times (though mixed layers may be deeper also). So understanding the drivers of the fluxes in each region is probably more important than a residence time calculated from a single surface measurement.

**Numerous missing references to previous work in the Atlantic:**

Not sure if there was some policy by the authors not to include pre-GEOTRACES work on Al in their discussion but there are several papers of direct relevance to this work that need to be included in the discussion as they directly address some of the questions the authors raised. In particular data on surface Al concentrations for dissolved (Gelado-Caballero et al., 1996; Helmers and van der loeff, 1993; Hydes, 1983; Kramer et al., 2004; Kremling, 1985; Kremling and Hydes, 1988; Moran and Moore, 1988; Moran and Moore, 1989; Sarthou et al., 2007) and particulate phases (Helmers, 1996; Kremling and Streu, 1993; Kuss and Kremling, 1999b; Moran and Moore, 1988; Moran and Moore, 1992; Wallace et al., 1981) along with data on the wet deposition of Al (Helmers and Schrems, 1995) and Al flux estimates from atmospheric concentrations (Jickells et al., 1994; Jickells, 1999; Powell et al., 2015). I am unaware of any analytical reason to exclude these data and the same analytical techniques are still used today.

**Analytical quality of the GA08 aluminium data and river discharge :**

The value of 784 nM that is reported from GA08 seems very doubtful unless some other information can be provided. Such a value is above the solubility limit for Al at seawater pH (May et al., 1979) and while it is close to undiluted river values for the Congo (Dupré et al., 1996; Meybeck, 1978; van Bennekom and Jager, 1978) most samples would be presumably located at least 12 miles offshore and thus significantly diluted. The linear range for most of the analytical systems is also not that large unless the sample is diluted prior to analysis. It raises questions then about the QA/QC applied to the data. If these samples were using the standard Lumogallion method (Hydes and Liss, 1976) as described in the methods section then they should have been corrected for the natural fluorescence of the samples as was pointed out previously for the Congo plume (van Bennekom and Jager, 1978). This correction should not be underestimated as the humic fluorescence at the excitation/emission used for Lumogallion can be considerable in humic rich waters. The methods that employ preconcentration schemes would not suffer from CDOM fluorescence.

At present the fluxes calculated for GA08 all seem to be too high because of the influence of the river plume and potentially the lack of a correction for CDOM fluorescence. The role of river inputs of Al could be compared to estimates of the riverine influence on the Atlantic (Cotrim da Cunha et al., 2007; Cotrim da Cunha et al., 2009) along with Al contents for the major rivers; e.g. Zaire river (Dupré et al., 1996; Meybeck, 1978; van Bennekom and Jager, 1978), Amazon, Orinoco (Mora et al., 2017) and Niger.

**Al composition of dust – the D term in the equation:**

The 8% value that Measures and Brown used in the original MADCOW was mentioned in the text but I could not find anywhere what value the authors decided to use (should be around 2.69 mmol/g-1 if 8% Al by weight and 26.981539 is the molecular weight for Al) and if they varied this according to region. If it is constant then the term could be incorporated into the S term to reduce the model variables. How valid is the assumption that it is constant? Could not some of the variability in the S term be related therefore to variation in the D term if other studies made the same assumption? At the very least the value used should be included somewhere in the text. Some explanation of how this was handled in the current work would be most illuminating!

**Specific Comments:**

P4 line 20: (sp) The chemical reagent is known as Lumogallion, not Lumogallium.

- P4 line 33. For consistency the dissolved Al concentrations should be in  $\mu$ mol m-3.
- P5 line 1. There is no explanation of what value is used for the D term in the equation. The other terms are explained in sections 2.2.1 2.2.3 but not the D term. If it is constant it could be included then in the S term.
- P5 line 7. This is a very large value  $\Delta\sigma_{\theta} = 0.125$  kg m-3 to use for determining the mixed layer depth as more recent work have shown that using smaller constraints  $\Delta\sigma_{\theta} = 0.03$  coupled with  $\Delta T = 0.2^{\circ}$  C provides a better estimate (de Boyer Montégut et al., 2004), this is in fact the threshold that is used in the Argo mixed layer climatology as cited in Holte et al. (2017). Thus it would be beneficial if the same criteria was used for the observed mixed layer depths to have a consistent approach. The problem with using a value  $\Delta\sigma_{\theta} = 0.125$  kg m-3 is that can seriously overestimate the mixed layer depth in high latitude areas leading to an increased inventory and longer residence time.
- P5 line 24. The authors should also be aware of work modelling the fractional solubility of aerosol Al (Han et al., 2012). It would therefore be prudent to include this work in the discussion and compare to the field data of Baker et al. (2013).
- P6 line 2. The residence time is a key variable in the version of MADCOW employed in this work and so it should be fairly well constrained. As the authors note the original version of MADCOW had the residence time fixed at 5 years along with the fractional solubility at 8% in order to simplify the calculations as changing one would impact the other. In the current approach it should be noted that the Han et al. (2008) work also includes many of the works that were not included in the citation list (see the general comments above) and these works were used to inform the residence times. It is also worth pointing out to the reader that Han et al. (2008) used a fixed mixed layer depth of 50 m and a constant solubility of 5% so this needs to be directly stated in the current manuscript with regard to how the values might compare.
- P6 line 10. The modelled residence times will include advection and mixing to an extent, but the use of a fixed solubility and mixed layer depth will also induce some key differences for the regions examined in the present work. This likely explains why the residence times are longer in the Han et al. (2008) work than in others as for many locations, the

underestimation of the solubility and the overestimation of the mixed layer will both work to increase the estimated residence time.

- P7 line 9. See the general comment above regarding this extremely high value of dissolved Al.
- P8 line 12. There is a considerable amount of surface data for this region and compiling it all in one place may reveal more about the seasonal timings of the dust flux to this region and the aluminium response. See the general comment above regards other works that have data for this region.
- P8 line 21. Not all of these studies attribute it to wet deposition, as the ITCZ acts partially as a barrier to the transport of the dust so the highest values are typically associated with direct dust deposition (Ravelo-Pérez et al., 2016; Tsamalis et al., 2013). Though precipitation is enhanced along the boundary between the ITCZ and the Saharan air layer (SAL) (Wilcox et al., 2010).
- P9 line 2. From where does the Al rich upwelled waters come from? Al profiles normally decrease with depth (scavenged profile) so this needs to be explained further as it would have then be more likely to be resuspension of Al rich particles close to the shelf rather than a direct upwelling source.
- P9 line 3. Do you mean an increased number of particles or that they were *enhanced* in some other fashion? Larger? More sticky?
- P9 line 4. See the general comment on this above.
- P9 line 8. (sp) reported
- P10 line 2. A strong control of the fractional solubility is the relative humidity/hygroscopicity of the particle as this controls the pH, aerosol acidity (Keene et al., 2002).
- P10 lines 22 and 24. This isn't a calculated result though, it is an estimate from a comparison with the work of Baker and colleagues.
- P10 line 27. See the general comment about relating the fractional solubility to the precipitation or relative humidity levels in the atmosphere for these regions.
- P11 line 8. It would be useful to see a plot of the residence times (as a 2D map or property-property plot) to see how they look on spatial scales and in relation to primary productivity if it is the main loss term for Al in the mixed layer.
- P11 line 33. It should be pointed out that statistically there are no differences between the values estimated here and those by Mahowald et al. (2005). So speculation on why the Mahowald is over estimated is somewhat spurious.
- P13 line 25. The more northerly flux values are likely underestimated as the residence time used is too long as it is likely in reality, days to weeks (see discussion about this above). This is an important point as the MADCOW model should work well where the Al fluxes and concentrations are the highest.
- P14 line 2. Most likely it clearly overestimates the fluxes when most of the Al is from the river.
- P14 line 12. See the general comment above about relating the seasonality of the dust fluxes.

- P14 line 25. So how do samples collected in the Pacific and Indian oceans tell us anything about deposition to the South Atlantic? Please explain this sentence more clearly.
- P15 line 12. ...lack of an island site...
- P15 line 16. It is great that Al is measured on GEOTRACES cruises but this does not make this approach using MADCOW any stronger as the majority of the development of this type of work was done pre-GEOTRACES.
- P15 line 21. Which IDP 2014 or 2017 both are citeable now.
- P15 line 23. For the Atlantic there are a number of north-south transects for Al and so some sort of seasonal signal is probably already possible and should be examined in the current work.
- Figure 2: Please state in the caption the climatology range used here, is it over an annual cycle?
- Table S5: The residence times used in this study are significantly shorter than what has been used previously in the MADCOW model (see above) and they are now on the same time scale as seasonal phytoplankton turnover so does this mean the residence time for Al can be scaled to productivity rather than input fluxes?
- Figure S4: The figure and the legend for this figure don't match up and there is no explanation of what the circles represent. While it is easy enough to conclude that the circles may represent discrete measurements at stations, the contoured data isn't explained and clearly does not share the same colour scale as the circles as the lowest value on the colour scale is blue and there is no blue in the contoured data. This figure needs to be fixed and explained better prior to acceptance.

**References cited:**

- Baker, A.R., Adams, C., Bell, T.G., Jickells, T.D. and Ganzeveld, L., 2013. Estimation of atmospheric nutrient inputs to the Atlantic Ocean from 50°N to 50°S based on large-scale field sampling: Iron and other dust-associated elements. Global Biogeochemical Cycles, 27(3): 755-767.
- Braun, S.A., 2010. Reevaluating the Role of the Saharan Air Layer in Atlantic Tropical Cyclogenesis and Evolution. Monthly Weather Review, 138(6): 2007-2037.
- Chester, R., 1982. Particulate Aluminium Fluxes in the Eastern Atlantic. Marine Chemistry, 11: 1-16.
- Cotrim da Cunha, L., Buitenhuis, E.T., Le Quéré, C., Giraud, X. and Ludwig, W., 2007. Potential impact of changes in river nutrient supply on global ocean biogeochemistry. Global Biogeochemical Cycles, 21(4).
- Cotrim da Cunha, L., Croot, P. and LaRoche, J., 2009. Influence of river discharge in the tropical and subtropical North Atlantic Ocean. Limnology and Oceanography, 54(2): 644-648.
- Croot, P.L., Streu, P. and Baker, A.R., 2004. Short residence time for iron in surface seawater impacted by atmospheric dry deposition from Saharan dust events. Geophysical Research Letters, 31: L23S08, doi:10.1029/2004GL020153.
- Dammshäuser, A., 2012. Distribution and behavior of the lithogenic tracers aluminium and titanium in the upper water column of the Atlantic Ocean, Christian-Albrechts-Universität Kiel, Kiel, Germany, 107 pp.
- Dammshäuser, A. and Croot, P.L., 2012. Low colloidal associations of aluminium and titanium in surface waters of the tropical Atlantic. Geochimica et Cosmochimica Acta, 96(0): 304-318.
- Dansie, A.P., Thomas, D.S.G., Wiggs, G.F.S. and Munkittrick, K.R., 2018. Spatial variability of ocean fertilizing nutrients in the dust-emitting ephemeral river catchments of Namibia. Earth Surface Processes and Landforms, 43(3): 563-578.
- Dansie, A.P., Wiggs, G.F.S. and Thomas, D.S.G., 2017a. Iron and nutrient content of wind-erodible sediment in the ephemeral river valleys of Namibia. Geomorphology, 290: 335-346.
- Dansie, A.P., Wiggs, G.F.S., Thomas, D.S.G. and Washington, R., 2017b. Measurements of windblown dust characteristics and ocean fertilization potential: The ephemeral river valleys of Namibia. Aeolian Research, 29: 30-41.
- de Boyer Montégut, C., Madec, G., Fischer, A.S., Lazar, A. and Iudicone, D., 2004. Mixed layer depth over the global ocean: An examination of profile data and a profile-based climatology. Journal of Geophysical Research: Oceans, 109(C12): n/a-n/a.
- Dupré, B., Gaillardet, J., Rousseau, D. and Allègre, C.J., 1996. Major and trace elements of riverborne material: The Congo Basin. Geochimica et Cosmochimica Acta, 60(8): 1301-1321.
- Gelado-Caballero, M.D., Torres-Padron, M.E., Hernandez-Brito, J.J., Herrera-Melian, J.A. and Perez-Pena, J., 1996. Aluminium distributions in Central East Atlantic waters (Canary Islands). Marine Chemistry, 51(4): 359-372.
- Han, Q., Moore, J.K., Zender, C., Measures, C. and Hydes, D., 2008. Constraining oceanic dust deposition using surface ocean dissolved Al. Global Biogeochemical Cycles, 22(2).
- Han, Q. et al., 2012. Global estimates of mineral dust aerosol iron and aluminum solubility that account for particle size using diffusion-controlled and surface-area-controlled approximations. Global Biogeochem. Cycles, 26(2): GB2038.
- Heimburger, A., Losno, R. and Triquet, S., 2013. Solubility of iron and other trace elements in rainwater collected on the Kerguelen Islands (South Indian Ocean). Biogeosciences, 10(10): 6617-6628.
- Helmers, E., 1996. Trace metals in suspended particulate matter of Atlantic Ocean surface water (40N to 20S). Marine Chemistry, 53: 51-67.
- Helmers, E. and Schrems, O., 1995. Wet Deposition of Metals to the Tropical North and the South-Atlantic Ocean. Atmospheric Environment, 29(18): 2475-2484.

- Helmers, E. and van der loeff, M.M.R., 1993. Lead And Aluminum In Atlantic Surface Waters (50-Degrees-N To 50-Degrees-S) Reflecting Anthropogenic And Natural Sources In The Eolian Transport. Journal Of Geophysical Research-Oceans, 98(C11): 20261-20273.
- Holte, J., Talley, L.D., Gilson, J. and Roemmich, D., 2017. An Argo mixed layer climatology and database. Geophysical Research Letters, 44(11): 5618-5626.
- Hwang, J., M. Druffel, E.R. and Eglinton, T.I., 2010. Widespread influence of resuspended sediments on oceanic particulate organic carbon: Insights from radiocarbon and aluminum contents in sinking particles. Global Biogeochemical Cycles, 24(4).
- Hwang, J., Manganini, S.J., Montluçon, D.B. and Eglinton, T.I., 2009. Dynamics of particle export on the Northwest Atlantic margin. Deep Sea Research Part I: Oceanographic Research Papers, 56(10): 1792-1803.
- Hydes, D.J., 1983. Distribution of aluminium in waters of the North East Atlantic 25°N to 35°N. Geochimica et Cosmochimica Acta, 47(5): 967-973.
- Hydes, D.J. and Liss, P.S., 1976. Fluorimetric method for determination of low concentrations of dissolved aluminum in natural waters. Analyst, 101: 922-931.
- Jickells, T., Church, T., Veron, A. and Arimoto, R., 1994. Atmospheric Inputs Of Manganese And Aluminum To The Sargasso Sea And Their Relation To Surface-Water Concentrations. Marine Chemistry, 46(3): 283-292.
- Jickells, T.D., 1999. The inputs of dust derived elements to the Sargasso Sea; a synthesis. Marine Chemistry, 68: 5-14.
- Jickells, T.D., Deuser, W.G. and Knap, A.H., 1984. The sedimentation rates of trace elements in the Sargasso Sea measured by sediment trap. Deep Sea Research Part A. Oceanographic Research Papers, 31(10): 1169.
- Keene, W.C., Pszenny, A.A.P., Maben, J.R. and Sander, R., 2002. Variation of marine aerosol acidity with particle size. Geophysical Research Letters, 29(7): 5-1-5-4.
- Kramer, J., Laan, P., Sarthou, G., Timmermans, K.R. and de Baar, H.J.W., 2004. Distribution of dissolved aluminium in the high atmospheric input region of the subtropical waters of the North Atlantic Ocean. Marine Chemistry, 88(3-4): 85-101.
- Kremling, K., 1985. The distribution of cadmium, copper, nickel, manganese, and aluminium in surface waters of the open Atlantic and European shelf area. Deep Sea Research Part A. Oceanographic Research Papers, 32(5): 531-555.
- Kremling, K. and Hydes, D., 1988. Summer distribution of dissolved Al, Cd, Co, Cu, Mn and Ni in surface waters around the British Isles. Continental Shelf Research, 8(1): 89-105.
- Kremling, K. and Streu, P., 1993. Saharan dust influenced trace element fluxes in deep North Atlantic subtropical waters. Deep-Sea Research, 40: 1155-1168.
- Kuss, J. and Kremling, K., 1999a. Particulate trace element fluxes in the deep northeast Atlantic Ocean. Deep-Sea Research, 46: 149-169.
- Kuss, J. and Kremling, K., 1999b. Spatial variability of particle associated trace elements in nearsurface waters of the North Atlantic (30°N/60°W to 60°N/2°W), derived by large volume sampling. Marine Chemistry, 68: 71-86.
- Kuss, J., Waniek, J.J., Kremling, K. and Schulz-Bull, D.E., 2010. Seasonality of particle-associated trace element fluxes in the deep northeast Atlantic Ocean. Deep Sea Research Part I: Oceanographic Research Papers, 57(6): 785-796.
- Liu, Z., Ostrenga, D., Teng, W. and Kempler, S., 2012. Tropical Rainfall Measuring Mission (TRMM) Precipitation Data and Services for Research and Applications. Bulletin of the American Meteorological Society, 93(9): 1317-1325.
- Losno, R. et al., 1993. Aluminum Solubility in Rainwater and Molten Snow. Journal of Atmospheric Chemistry, 17(1): 29-43.
- May, H.M., Helmke, P.A. and Jackson, M.L., 1979. Gibbsite Solubility And Thermodynamic Properties Of Hydroxy-Aluminum Ions In Aqueous-Solution At 25-Degrees-C. Geochimica Et Cosmochimica Acta, 43(6): 861-868.

- Meybeck, M., 1978. Note On Dissolved Elemental Contents Of The Zaire River. Netherlands Journal Of Sea Research, 12(3-4): 293-295.
- Mora, A. et al., 2017. Dynamics of dissolved major (Na, K, Ca, Mg, and Si) and trace (Al, Fe, Mn, Zn, Cu, and Cr) elements along the lower Orinoco River. Hydrological Processes, 31(3): 597-611.
- Moran, S.B. and Moore, R.M., 1988. Temporal variations in dissolved and particulate aluminium during a spring bloom. Estuarine, Coastal and Shelf Science, 27: 205-215.
- Moran, S.B. and Moore, R.M., 1989. The distribution of colloidal aluminum and organic carbon in coastal and open ocean waters off Nova Scotia. Geochimica et Cosmochimica Acta, 53: 2519-2527.
- Moran, S.B. and Moore, R.M., 1991. The Potential Source Of Dissolved Aluminum From Resuspended Sediments To The North-Atlantic Deep-Water. Geochimica Et Cosmochimica Acta, 55(10): 2745-2751.
- Moran, S.B. and Moore, R.M., 1992. Kinetics of the removal of dissolved aluminium by diatoms in seawater: A comparison with thorium. Geochimica et Cosmochimica Acta, 56: 3365-3374.
- Pohl, C. et al., 2011. Synoptic transects on the distribution of trace elements (Hg, Pb, Cd, Cu, Ni, Zn, Co, Mn, Fe, and Al) in surface waters of the Northern- and Southern East Atlantic. Journal of Marine Systems, 84(1-2): 28-41.
- Powell, C.F. et al., 2015. Estimation of the Atmospheric Flux of Nutrients and Trace Metals to the Eastern Tropical North Atlantic Ocean\*. Journal of the Atmospheric Sciences, 72(10): 4029-4045.
- Ravelo-Pérez, L.M. et al., 2016. Soluble iron dust export in the high altitude Saharan Air Layer. Atmospheric Environment, 133(Supplement C): 49-59.
- Sarthou, G. et al., 2007. Influence of atmospheric inputs on the iron distribution in the subtropical North-East Atlantic Ocean. Marine Chemistry, 104(3-4): 186-202.
- Tsamalis, C., Chédin, A., Pelon, J. and Capelle, V., 2013. The seasonal vertical distribution of the Saharan Air Layer and its modulation by the wind. Atmos. Chem. Phys., 13(22): 11235-11257.
- van Bennekom, A.J. and Jager, J.E., 1978. Dissolved Aluminum In The Zaire River Plume. Netherlands Journal Of Sea Research, 12(3-4): 358-367.
- Wallace, G.T., Mahoney, O.M., Dulmage, R., Storti, F. and Dudek, N., 1981. First-order removal of particulate aluminium in oceanic surface layers. Nature, 293(5835): 729-731.
- Wilcox, E.M., Lau, K.M. and Kim, K.M., 2010. A northward shift of the North Atlantic Ocean Intertropical Convergence Zone in response to summertime Saharan dust outbreaks. Geophysical Research Letters, 37(4).

---

## Referee Comment (RC1) · Anonymous Referee #1 · 31 May 2018

bg-2018-209 Submitted on 26 Apr 2018

Atmospheric deposition fluxes over the Atlantic Ocean: A GEOTRACES case study

This paper uses the MADCOW model to calculate dust fluxes to the Atlantic Ocean, and compares the results to a dust flux model from Mahowald.

The key issue in this comment is that the residence times used to calculate dust fluxes are obtained from Han et al 2008. Those residence times are calculated using the DEAD dust flux model and the BEC ocean circulation and biogeochemistry model for the dissolved Al distribution. The MADCOW model formulation is this:

[Figure]

G = ([Al]*MLD)/(T*S*D)

Where:

G=dust flux (grams per square meter per year)

[Al] = the dissolved Al concentration in the mixed layer (moles per cubic meter, NOT moles per liter!!)

MLD = mixed layer depth (meters)

T= residence time (years) (from Han et al., 2008)

S= fractional solubility

D= Al concentration in dust (moles/gram)

T is the [Al] inventory from the BEC model divided by the sum of the inputs of dissolved Al from dust and from mixing. The dissolved Al flux from dust was derived from the DEAD dust model using a solubility of 5% and 8% Al in dust (0.002965 moles/gram) and the mixing terms were obtained from the BEC model.

So, T can be written as:

T= $([Al]*MLD)_{BEC\ model}/(G*S*D+mixing)_{DEAD\ and\ BEC\ model}$

T= $([Al]*MLD)_{BEC\ model}/(G*0.05*0.002965 + mixing)_{DEAD\ and\ BEC\ model}$

Substituting T into the MADCOW equation yields:

$G_{Atlantic}$ = $([Al]*MLD_{Atlantic} *(G*0.05*0.002965 + mixing)_{DEAD\ and\ BEC\ model})/$

$([Al]*MLD)_{BEC\ model}*(S*D)_{Atlantic})$

When the mixing terms for the dissolved Al input (from the DEAD+BEC model) are small, we can further resolve this equation. I assume they both used 8% Al in dust (D=0.002965 moles/gram) so the D terms cancel.

$G_{Atlantic}/G_{DEAD}$ = ([Al]*MLD$_{Atlantic}$)/ [Al]*MLD)$_{BEC\ model}$)*((0.05)$_{DEAD}$/(S)$_{Atlantic}$)

This equation can therefore be used to calculate the ratio of the dust fluxes in this paper to those used by Han et al. (2008) from the DEAD model. You can see that the dust flux ratio is affected by the ratio of the dissolved Al inventory (from the Atlantic data in this paper) to the inventories for the same locations from the BEC model in Han et al. (2008) and, equally as important, by the ratio of the Al solubilities, where the DEAD model used a fixed value of 5% and this paper uses a variety of solubilities obtained from actual aerosol measurements across the Atlantic. If they used a different Al concentration in dust in this paper (it is not specified!) then the D terms would not cancel, further affecting the dust flux ratios.

If the dissolved Al inventories from this paper are the same as those obtained by Han et al. (2008) using the BEC model and if the same fractional solubility is used, then the dust fluxes would be the same and the dust flux ratio would be 1.0.

This paper (Table S3) uses Al solubilities always greater than or equal to 5% (often 2-3 times higher), so if the dissolved Al inventories in this paper and from the BEC model are similar, then the predicted dust flux would always be less than or equal to what the DEAD model shows (and probably also less than or equal to what the Mahowald dust model shows). This is a simple mathematical outcome; it does not really say anything substantially new about the MADCOW model and its ability to compare with dust flux models.

It would also be very instructive to compare the new dissolved Al inventories in this paper to the inventories for the same locations from the BEC model; if the BEC model

inventories are very different from those shown in this paper, then the dust fluxes would not agree with the DEAD model fluxes even if they used the same fractional solubility!

At the very least, using fractional Al solubilities from the Atlantic data in this paper to calculate dust fluxes that are then compared to the Mahowald dust model fluxes is not the correct comparison to make. The dust fluxes should be compared to the DEAD model dust fluxes, since those fluxes were used to estimate the residence times. And the degree of disagreement can then be attributed to differences in the dissolved Al inventories (measured vs. modeled) and/or differences in the Al fractional solubility. This makes the paper less "descriptive" and more "quantitative".

---

## Referee Comment (RC2) · Anonymous Referee #1 · 31 May 2018

bg-2018-209 Submitted on 26 Apr 2018 Atmospheric deposition fluxes over the Atlantic Ocean: A GEOTRACES case study

Minor typos and comments Page: Line: Comment:

3: 17: use particle collection

3: 27: The MADCOW model uses more than one parameter; the residence time is a derived or assumed value, and it is probably the least well know term in the equation.

4: 17: Please express the acidification of the samples with the molar concentration of acid added. For example, if you add 4 mL of 6M HCl per liter, you added 0.024M HCl.

[Figure]

That would have a pH around 1.7-1.8.

4: 31: The dissolved Al must be in moles per cubic meter units.

14: 9: enhanced

14: 11: likely results in

14: 26: and was somewhat lower

15: 10: constraints

15: 12: sites

15: 15: I would delete "which implies a major strength of the approach used in this study" because this study does not reveal anything substantially new about the use of dissolved Al in the MADCOW model.

15: 18: such as

15: 30: Special thanks

Figure 6: panel (b) needs coordinate values on the axes.

Table 1: Should compare to DEAD dust fluxes, not Mahowald.

Table S4: Add two columns to show the residence time and aerosol Al solubility used for the dust flux calculation for each station number.

Figure S5: This is the most useful figure and should be moved into the main body of the paper, where you could discuss why the calculated dust fluxes disagree or agree with the DEAD model fluxes. Is the disagreement due to differences between the observed and BEC-modeled dissolved Al inventories or because you used a higher fractional aerosol Al solubility?

---

## Referee Comment (RC3) · Anonymous Referee #2 · 8 Jun 2018

Review of Menzel Barraqueta Atmospheric supply of trace elements has been a central theme of GOETRACES and so this paper is an appropriate contribution to this issue. The paper attempts to use aluminium data in the water column to estimate atmospheric dust deposition in a refinement of the MADCOW model developed by Chris Measures and colleagues. The data and approaches involved are basically sound and I am happy to recommend publication but would suggest some modifications before publication. I have two general points. 1. These authors another paper submitted to this issue which is referenced here and which is partially repeated here. There is also a lot of information in the paper that notes the similarity of the data reported on aluminium concentrations to that previously reported. I cannot help feeling that much
of this material could be shortened in this paper if the focus of the paper is indeed on the utility of the MADCOW model. 2. The MADCOW model was always acknowledged to require assumptions about mixed layer depth, solubility and dAI scavenging. These are explored in detail here but firstly it should be clear that these limitations of the model have been acknowledged by the community for a long time. Secondly with at least these three parameters as numbers that, even with the careful regional evaluations here, are poorly known, there are limitations to how far the model can be used in a detailed area specific concentration mode. Specific points Line 23-24 I don't think that clouds compromise deposition flux estimates Line 12-20 There is no mention of filtration in the methods here - if the data were for unfiltered samples acidified in this way it would include much of the pAl. In the other submitted paper it says the samples were filtered which is I assume the case but this needs to be clarified. 2.2.3 The use of the Han residence time approaches seems appropriate but if the output is essentially that of Han the subsequent discussion of it could perhaps be shortened. 3.1 Mixed layer depth is a key component of the MADCOW model and clearly varies from place to place and from season to season. The discussion here emphasises the large resultant uncertainties but does not discuss how and why they arise or the best approach to dealing with them. It is not actually clear to me even which of the various MLD estimates were used. P7 section 3.2 is actually 3.3 I think. There is I think a lot of general review of other data throughout section 3.3 that seems to me could be shortened since it has been discussed in the cited papers and the dAl distribution in the Atlantic is quite well known. 3.2.1 line 23 what criteria are used to exclude continental input influenced data? Section 3.3. lines 8-10 and line 12 are contradictory. The different solubilisation methods do yield systematically different values but these difference can be accounted for and are not the main causes of the difficulties in estimating atmospheric deposition. Line 19-28 I am not sure that there is evidence for AI sources with very different solubilities in the way that has been shown to be important for anthoprogenic vs dust Fe sources. Atmospheric processing is important (line 26) as shown by Baker and Croot and Sholkovitch. P11 line 10 I would think Table S5 should be in the main paper given
its importance to the results. P13 Line 15. I wonder why the comparison is to the Duce et al 1991 paper when there are more recent maps for dust deposition at least. Line 25-30 the MADCOW model did not ever aspire to "accurately determine atmospheric deposition fluxes" P15 line 9 when the MADCOW and atmospheric dust deposition models diverge, it is not clear to me that it is possible to know which is right and wrong as implied here

---

## Author Comment (AC1) · 1 Oct 2018

Reviewer 1:

This paper uses the MADCOW model to calculate dust fluxes to the Atlantic Ocean, and compares the results to a dust flux model from Mahowald. The key issue in this comment is that the residence times used to calculate dust fluxes are obtained from Han et al 2008. Those residence times are calculated using the DEAD dust flux model and the BEC ocean circulation and biogeochemistry model for the dissolved Al distribution.

Indeed, our choice of Han et al., 2008 for the residence times was not arbitrary. We could have used our own estimated residence times. However, we would have fallen into a circular approach since we would have calculated atmospheric fluxes from calculated residence times using as input Mahowald deposition fluxes and later on compare our calculated fluxes against Mahowald fluxes. This would have been inappropriate.

The MADCOW model formulation is this:

G = ([AI]\*MLD)/(T\*S\*D)

Where:

G=dust flux (grams per square meter per year)

[AI] = the dissolved AI concentration in the mixed layer (moles per cubic meter, NOT

moles per liter!!)

→ This was a mistake and has been corrected

MLD = mixed layer depth (meters)

T= residence time (years) (from Han et al., 2008)

S= fractional solubility

D= Al concentration in dust (moles/gram)

T is the [AI] inventory from the BEC model divided by the sum of the inputs of dissolved

Al from dust and from mixing. The dissolved Al flux from dust was derived from the

DEAD dust model using a solubility of 5% and 8% Al in dust (0.002965 moles/gram)

and the mixing terms were obtained from the BEC model.

So, T can be written as:

T= ([AI]\*MLD)BEC model/(G\*S\*D+mixing)DEAD and BEC model

T= ([AI]\*MLD)BEC model/(G\*0.05\*0.002965 + mixing)DEAD and BEC model

Substituting T into the MADCOW equation yields:

GAtlantic = ([AI]\*MLDAtlantic \*(G\*0.05\*0.002965 + mixing)DEAD and BEC model)/

([AI]\*MLD)BEC model\*(S\*D)Atlantic)

When the mixing terms for the dissolved Al input (from the DEAD+BEC model) are small, we can further resolve this equation. I assume they both used 8% Al in dust (D=0.002965 moles/gram) so the D terms cancel. GAtlantic /GDEAD = ([AI]\*MLDAtlantic)/ [AI]\*MLD)BEC

model)\*((0.05)DEAD/(S)Atlantic) This equation can therefore be used to calculate the ratio of the dust fluxes in this paper to those used by Han et al. (2008) from the DEAD model. You can see that

the dust flux ratio is affected by the ratio of the dissolved Al inventory (from the Atlantic data in this paper) to the inventories for the same locations from the BEC model in Han et al. (2008) and, equally as important, by the ratio of the Al solubilities, where the DEAD model used a fixed value of 5% and this paper uses a variety of solubilities obtained from actual aerosol measurements across the Atlantic.

If they used a different Al concentration in dust in this paper (it is not specified!) then the D terms would not cancel, further affecting the dust flux ratios. If the dissolved Al inventories from this paper are the same as those obtained by Han et al. (2008) using the BEC model and if the same fractional solubility is used, then the dust fluxes would be the same and the dust flux ratio would be 1.0. This paper (Table S3) uses Al solubilities always greater than or equal to 5% (often 2-3 times higher), so if the dissolved Al inventories in this paper and from the BEC model are similar, then the predicted dust flux would always be less than or equal to what the DEAD model shows (and probably also less than or equal to what the Mahowald dust model shows). This is a simple mathematical outcome; it does not really say anything substantially new about the MADCOW model and its ability to compare with dust flux models. It would also be very instructive to compare the new dissolved Al inventories in this paper to the inventories for the same locations from the BEC model; if the BEC mode inventories are very different from those shown in this paper, then the dust fluxes would not agree with the DEAD model fluxes even if they used the same fractional solubility!

We did used the same value for D as in the original manuscript (8.1% Al in dust). We did calculate our dAl inventories by trapezoidal integration in order to calculate our own residence times. However, as we were using as input the Mahowald dust fluxes and we wanted to compare our calculated atmospheric fluxes again Mahowald fluxes we cancelled our residence times and used published values. We presume that our inventories will be different to the ones presented in Han et al. 2008 since the depth of the mixed layer differs between our study and Han et al. modelling manuscript. We do not have access to the inventories of the BEC model since we are not able to contact Qin Han as she left academia after her PhD.

At the very least, using fractional Al solubilities from the Atlantic data in this paper to calculate dust fluxes that are then compared to the Mahowald dust model fluxes is not the correct comparison to make. The dust fluxes should be compared to the DEAD model dust fluxes, since those fluxes were used to estimate the residence times. And the degree of disagreement can then be attributed to differences in the dissolved Al inventories (measured vs. modeled) and/or differences in the Al fractional solubility. This makes the paper less "descriptive" and more "quantitative"

We have added DEAD model dust fluxes. As Han has left science, the data appears challenging to track down, but other leads are followed.

Minor typos and comments Page: Line: Comment:

3: 17: use particle collection

Done

3: 27: The MADCOW model uses more than one parameter; the residence time is a derived or assumed value, and it is probably the least well know term in the equation.

We have removed that sentence to avoid confusion

4: 17: Please express the acidification of the samples with the molar concentration of acid added. For example, if you add 4 mL of 6M HCl per liter, you added 0.024M HCl.

That would have a pH around 1.7-1.8.

Done

4: 31: The dissolved Al must be in moles per cubic meter units.

Done

14: 9: enhanced

Done

14: 11: likely results in

Done

14: 26: and was somewhat lower

Done

15: 10: constraints

Done

15: 12: sites

Done

15: 15: I would delete "which implies a major strength of the approach used in this

study" because this study does not reveal anything substantially new about the use of

dissolved Al in the MADCOW model.

Modified to: " which implies a major strength of the MADCOW model"

15: 18: such as

Done

15: 30: Special thanks

Done

Figure 6: panel (b) needs coordinate values on the axes.

Done

Table 1: Should compare to DEAD dust fluxes, not Mahowald.

DEAD fluxes added. We will also keep the comparison against Mahowald fluxes since this provides us with additional insights into the differences between the various dust deposition approaches.

Table S4: Add two columns to show the residence time and aerosol Al solubility used

for the dust flux calculation for each station number.

Done

Figure S5: This is the most useful figure and should be moved into the main body of the paper, where you could discuss why the calculated dust fluxes disagree or agree with the DEAD model fluxes. Is the disagreement due to differences between the observed

and BEC-modeled dissolved Al inventories or because you used a higher fractional

aerosol Al solubility?

*We have moved the figure to the main body. Now Figure 7. We have renumbered all other figures accordingly to the change.*

We are not able to compare inventories, but they chances they are equal are minimal. Both factors are different and presumably disagreements at any certain station are a consequence of variability within the latter factors.

---

## Author Comment (AC2) · 1 Oct 2018

Dear Editor and referee 1,

The answers for RC2 are attached in the supplement file for RC1 as it was the same referee. After each referee comment you can find our answer in cursive font.

Best regards,

Jan-Lukas Menzel Barraqueta and co-authors

---

## Author Comment (AC3) · 1 Oct 2018

Reviewer 2:

Review of Menzel Barraqueta Atmospheric supply of trace elements has been a central theme of GOETRACES and so this paper is an appropriate contribution to this issue. The paper attempts to use aluminium data in the water column to estimate atmospheric dust deposition in a refinement of the MADCOW model developed by Chris Measures and colleagues. The data and approaches involved are basically sound and I am happy to recommend publication but would suggest some modifications before publication. I have two general points.

1. These authors another paper submitted to this issue which is referenced here and which is partially repeated here. There is also a lot of information in the paper that notes the similarity of the data reported on aluminium concentrations to that previously reported. I cannot help feeling that much of this material could be shortened in this paper if the focus of the paper is indeed on the utility of the MADCOW model.

*Indeed, the dissolved aluminium data from GEOTRACES section GA01 has been published in a different manuscript in the special issue. However, in this manuscript we are describing the dissolved aluminium signature within the mixed layer depth and as such it varies in comparison with the other manuscript. Also, in order to understand and explain the MADCOW model outputs it is necessary to describe the dAl signature within the mixed layer depth. We have attempted to keep the discussion of dAl as brief as possible, but were requested by reviewer 3 to add some further references and text to explain the geographical variations.*

2. The MADCOW model was always acknowledged to require assumptions about mixed layer depth, solubility and dAl scavenging. These are explored in detail here but firstly it should be clear that these limitations of the model have been acknowledged by the community for a long time.

Secondly with at least these three parameters as numbers that, even with the careful regional evaluations here, are poorly known, there are limitations to how far the model can be used in a detailed area specific concentration mode.

*We acknowledged the comment by the reviewer, and indeed explore these limitations in the manuscript. Reviewer 3 makes a similar comment.*

Specific points

Line 23-24 I don't think that clouds compromise deposition flux estimates

*Clouds itself do not compromise deposition fluxes. However, deposition fluxes derived from satellite derived climatologies often are biased to clear sky conditions. Aerosol optical depth properties suffer then from cloud presence.*

*We have reformulated the sentence as follow:*

*Modelled atmospheric deposition fluxes rely on satellite-derived climatologies. The latter climatologies use properties (i.e aerosol optical depth) which suffer from interferences from cloud coverage and are biased towards clear sky conditions (Huneeus et al., 2011).*

Line 12-20. There is no mention of filtration in the methods here – if the data were for unfiltered samples acidified in this way it would include much of the pAl. In the other submitted paper it says the samples were filtered which is I assume the case but this needs to be clarified.

*Yes, all the samples were filtered. In table S2 you can find the filter type and pore size. We now mentioned it in the main text (section 2.1.)*

2.2.3 The use of the Han residence time approaches seems appropriate but if the output is essentially that of Han the subsequent discussion of it could perhaps be shortened.

*We feel that the subsequent discussion is needed in order to provide background information on the variability of the residence time regarding different oceanic regions.*

*We have shortened the section.*

3.1 Mixed layer depth is a key component of the MADCOW model and clearly varies from place to place and from season to season. The discussion here emphasises the large resultant uncertainties but does not discuss how and why they arise or the best approach to dealing with them. It is not actually clear to me even which of the various MLD estimates were used.

*We acknowledge your comment. We do acknowledge the factors that drive changes in the depth of the mixed layer and which ones do play a major role within each area. The best approach would be to assess values on a station per station basis. However, this would difficult the intra-comparison of dust fluxes within the same cruise.*

*As input parameter for the MADCOW model we have chosen to use a single mixed layer depth value for each cruise. This single value is the median value of the in situ MLD and the annual MLD from the Argo project.*

*We now explicit acknowledge the value used in the text.*

*"As input parameter for the MADCOW model we have chosen a single MLD value for each cruise. The latter is the median value between the MLDms and MLDar. We acknowledge that this may not be the best approach but it gives us the opportunity for intra comparison of atmospheric fluxes within the same cruise"*

P7 section 3.2 is actually 3.3 I think. There is I think a lot of general review of other data throughout section 3.3 that seems to me could be shortened since it has been discussed in the cited papers and the dAl distribution in the Atlantic is quite well known.

*Indeed, this is a mistake from our side. It is section 3.2. The following subsections have been re-numbered accordingly (3.2.1, 3.2.2, 3.2.3, 3.2.4).*

*We have shortened the section regarding GA01. However, dAl data for GA06, GA08, and GA10 are new and need to be discussed and compared with previous data.*

3.2.1 line 23 what criteria are used to exclude continental input influenced data?

*It is written some lines above. Normally, background concentrations are used. The stations excluded are all "coastal stations". In the previous manuscript dealing with the GA01 dataset (Menzel Barraqueta et al., 2018) we explained the different sources which could have increased the dAl levels in these waters.*

Section 3.3. lines 8-10 and line 12 are contradictory. The different solubilisation methods do yield systematically different values but these difference can be accounted for and are not the main causes of the difficulties in estimating atmospheric deposition.

*We acknowledge your comment. Indeed, the different leaching methods do yield different results due to difference pH of leach media, longer exposure time to HAc leach than UHP water leach, different ionic concentrations of leach media etc. Results should not be extrapolated from one method to another method. However, the GEOTRACES data suggests that there is roughly a tenfold increase in solubility of aerosol Al from samples leached with HAc compared to UHP water. You are right, the main difficulty in estimating atmospheric deposition from aerosol concentrations remain in the large uncertainty in deposition velocities and in extrapolating a snap shot measurement into an annual deposition value.*

Line 19-28 I am not sure that there is evidence for Al sources with very different solubilities in the way that has been shown to be important for anthoprogenic vs dust Fe sources. Atmospheric processing is important (line 26) as shown by Baker and Croot and Sholkovitch.

*We acknowledge your comment. Indeed, atmospheric processing during transport is an important factor.  However, it has been demonstrated that aerosols from different sources and from different nature do show different solubilities (Baker and Jickells, 2017, Baker et al., 2013, Baker et al., 2006).*

 P11 line 10 I would think Table S5 should be in the main paper given its importance to the results.

*As suggested, we have moved Table S5 to the main paper. Now it is Table 1 and the original Table 1 has been change to Table 2.*

-P13 Line 15. I wonder why the comparison is to the Duce et al 1991 paper when there are more recent maps for dust deposition at least.

*Our main comparison is against Mahowald et al., 2005. We have included Duce et al., 1991 as additional information and because it was one of the first global ocean maps for atmospheric deposition. Following to comments of reviewer 1, we have added atmospheric fluxes derived from the DEAD model.*

Line 25-30 the MADCOW model did not ever aspire to "accurately determine atmospheric deposition fluxes"

*We have modified the sentence as follow:*

*These results do not match the observations (from field data and satellite retrievals) and suggests that atmospheric deposition fluxes calculated with the MADCOW model are less reliable in the tropical North Atlantic Ocean…….*

P15 line 9 when the MADCOW and atmospheric dust deposition models diverge, it is not clear to me that it is possible to know which is right and wrong as implied here

*You are right. It is not possible to know which one is correct. We have rewritten the sentence to avoid confusion.*

*"Our atmospheric deposition fluxes were lower than model fluxes in areas of the Atlantic Ocean regions removed from the main aerosol sources regions. This observation suggests that these regions receive less atmospheric inputs than the models indicate or that MADCOW underestimates atmospheric inputs to these regions."*

---

## Author Comment (AC4) · 1 Oct 2018

Reviewer 3:

Overview:

This manuscript presents results of the application of the MADCOW model for aerosol deposition to recent GEOTRACES data from the Atlantic. The authors expand on the original MADCOW model by varying previously fixed parameters through a combination of comparison with field data for fractional solubility and model date for residence times. While it is an interesting topic, much of the discussion reads like a summary of the earlier works and the manuscript would be better focused on providing new insights into the GEOTRACES datasets through examining how well the assumptions in the MADCOW model are adhered to. There are some question marks regarding the GA08 Al data set also as it the dissolved Al values appear to be overestimated possibly due to the lack of correction for CDOM fluorescence due to the methodology that was used during that expedition. Overall this paper does a good job in adding value to existing GEOTRACES datasets and could make a very useful contribution to this field if it is revised along the lines outlined below.

General Comments:

Atmospheric fluxes – wet and dry

While the paper does a reasonable job of explaining how the fluxes were calculated it does not get into a detailed comparison with atmospheric based fluxes for which there is also data from GEOTRACES and other programs. One aspect of the current work where the atmospheric data would help decipher things is in assigning how much of the surface Al comes from aerosol flux (dry deposition) and how much from wet deposition. In this regard making the link to the precipitation fluxes for each region (Liu et al., 2012) would be beneficial in examining if this is what determine the high inferred model solubility of the aerosols or not. As the assumption of the MADCOW model is that dry deposition is the only process occurring and that in areas where wet deposition is important a higher fractional solubility is assumed. There are data for aluminium solubility in marine rain (Heimburger et al., 2013; Losno et al., 1993), the Losno et al. (1993) paper includes several samples from the Atlantic. See also for example the impact of the Saharan air layer and the ITCZ on the relative humidity in the atmosphere (Braun, 2010). Addition of this type of analysis would greatly increase the impact of this work.

*We now have commented on the differentiation between dry and wet deposition influence on Al solubility. We are not really sure what you mean when saying "if this is what determine the high inferred model solubility of the aerosols or not". In our model? If this is the case, the answer is no. The Al solubility values we use are all inferred from dry atmospheric deposition. However, it is true that the humidity present within different air masses coupled with large range transport of aerosols may play an important role affecting Al fractional solubility. The latter is valid also for dry deposition after a long transport within humid air mass layers.*

*We do not understand what you mean with atmospheric based fluxes. In case it is aerosol and rain concentration data, we do make comparisons of our calculated deposition fluxes against them. We also have tried to compare against other tracers as for example [7]Be based deposition fluxes. However, the main goal was to compare our calculated fluxes against modelling fluxes since most of our data (mainly apart of cruise GA06) are from remote areas with few or any discrete deposition fluxes published. By comparing against Mahowald model we could extract the atmospheric flux for the same location as our calculated flux and therefore we are able to compare them. We now also compared our fluxes against DEAD model fluxes as suggested by reviewer 1.*

*We are not sure if you write about the parameters we used to constrain the MADCOW model in our study or if you refer to the actual original MADCOW model.*

Seasonality and residence time:

A critical weakness of simple box models like the MADCOW model is that areas with strong seasonality of inputs/outputs are inadequately described when using a single concentration term to fix the inventory. Previous work has indicated that the seasonal cycle (or interannual variability) off the west African coast is on the order of 60 nM for dissolved Al (Pohl et al., 2011) and presumably the residence time is then shorter than 2-5 years first postulated by Helmers and van der loeff (1993). Indeed comparison with Fe suggests that the residence time could be much less than a year or so (Croot et al., 2004; Dammshäuser, 2012; Dammshäuser and Croot, 2012) in these high dust impacted regions. At present there is little discussion regarding the assumptions inherent in a steady state model such as MADCOW, the focus in the paper is on the inventory size as determined by mixed layer depth and concentration and not on whether the fluxes are in balance over the time scales being investigated.

In this regard there are a number of studies that have looked at the seasonality of particle fluxes of Al in the North Atlantic (Chester, 1982; Hwang et al., 2010; Hwang et al., 2009; Jickells, 1999; Jickells et al., 1984; Kuss and Kremling, 1999a; Kuss et al., 2010). With regard to the seasonality in the Benguela region, there has been recent work looking at the fluxes from the Namib (Dansie et al., 2018; Dansie et al., 2017a; Dansie et al., 2017b) and their predominance during austral winter that is of relevance here to the question of inputs and residence times. The challenge that arises then is how to reconcile a snap shot residence time provided by a single concentration measurement within a very active seasonal cycle. For example most sampling is in summer which while likely to be the maximum sink for dissolved Al due to enhanced biological productivity and scavenging, but also could be a minimum in atmospheric deposition leading to a residence time of weeks. Contrastingly winter measurements may have higher deposition rates and minimal scavenging resulting in longer apparent residence times (though mixed layers may be deeper also). So understanding the drivers of the fluxes in each region is probably more important than a residence time calculated from a single surface measurement.

*This is a very interesting and good point. We now have added sentences explicitly addressing this issue. There is seasonality and this is now acknowledged in the text. In the tropical Atlantic seasonality plays a major role, especially through changes in the position of the ITCZ and the nature of episodic dust deposition events.*

*The point raised by the reviewer is quite tricky and more work is needed in the future to address this issues. Also, an important point made recently is the dual role of aerosols as a sink and a source of trace metals (Ye and Volker, 2017).*

*The study of Pohl et al., 2011 makes a great effort in examining the distribution of trace metals along a North to South transect and comparing it against a previous transect in 1990. However, in the latter study no dissolved Al samples were taken and only total Al was analysed. As such, the data (dissolved Al against total Al) are not directly comparable and any comparison would be merely speculative. We are not able to find the seasonal variation of 60 nM total (dissolved mentioned in the comment) Al in the manuscript mentioned in your comment.*

*We have added a sentence acknowledging the effort made by different colleagues on the limitations and assumptions of the MADCOW model. "The limitations of the MADCOW model and extended discussions on the inherent assumptions of the MADCOW model have been acknowledge in previous investigations (e.g Measures and Brown, 1996; Measures and Vink, 2000)..*

Numerous missing references to previous work in the Atlantic:

Not sure if there was some policy by the authors not to include pre-GEOTRACES work on Al in their discussion but there are several papers of direct relevance to this work that need to be included in the discussion as they directly address some of the questions the authors raised. In particular data on surface Al concentrations for dissolved (Gelado-Caballero et al., 1996; Helmers and van der loeff, 1993; Hydes, 1983; Kramer et al., 2004; Kremling, 1985; Kremling and Hydes, 1988; Moran and Moore, 1988; Moran and Moore, 1989; Sarthou et al., 2007) and particulate phases (Helmers, 1996; Kremling and Streu, 1993; Kuss and Kremling, 1999b; Moran and Moore, 1988; Moran and Moore, 1991; Moran and Moore, 1992; Wallace et al., 1981) along with data on the wet deposition of Al (Helmers and Schrems, 1995) and Al flux estimates from atmospheric concentrations (Jickells et al., 1994; Jickells, 1999; ). I am unaware of any analytical reason to exclude these data and the same analytical techniques are still used today.

*We acknowledge your comment. There was no policy to not include pre-GEOTRACES work as some pre-GEOTRACES work has been included (e.g. Vink and Measures., 2001, Van der Loeff et al., 1997, Bowie et al., 2002, Measures and Vink., 2000, Van Bennekom and Jager 1978). In our previous work (same issue) we nearly cite all the dAl works you mentioned above.* We now have added more references of dAl data within the text and *we have included the works for Al flux estimates from atmospheric concentration data (Jickells, 1999 was already included).*

*One of the main and powerful tools of GEOTRACES is the need of running reference material which was not done in the pre-GEOTRACES era. This does not mean that data before the GEOTRACES are not of high quality but they have not been (or only occasionally) cross check against reference seawater or inter-calibrated.*

Analytical quality of the GA08 aluminium data and river discharge:

The value of 784 nM that is reported from GA08 seems very doubtful unless some other information can be provided. Such a value is above the solubility limit for Al at seawater pH (May et al., 1979) and while it is close to undiluted river values for the Congo (Dupré et al., 1996; Meybeck, 1978; van Bennekom and Jager, 1978) most samples would be presumably located at least 12 miles offshore and thus significantly diluted. The linear range for most of the analytical systems is also not that large unless the sample is diluted prior to analysis. It raises questions then about the QA/QC applied to the data. If these samples were using the standard Lumogallion method (Hydes and Liss, 1976) as described in the methods section then they should have been corrected for the natural fluorescence of the samples as was pointed out previously for the Congo plume (van Bennekom and Jager, 1978). This correction should not be underestimated as the humic fluorescence at the excitation/emission used for Lumogallion can be considerable in humic rich waters. The methods that employ preconcentration schemes would not suffer from CDOM fluorescence. At present the fluxes calculated for GA08 all seem to be too high because of the influence of the river plume and potentially the lack of a correction for CDOM fluorescence. The role of river inputs of Al could be compared to estimates of the riverine influence on the Atlantic (Cotrim da Cunha et al., 2007; Cotrim da Cunha et al., 2009) along with Al contents for the major rivers; e.g. Zaire river (Dupré et al., 1996; Meybeck, 1978; van Bennekom and Jager, 1978), Amazon, Orinoco (Mora et al., 2017) and Niger.

*We did correct for natural fluorescence as stated on the original method of Hydes and Liss. Buffered sample. We also diluted several times the samples within the Congo River plume. This high value of dAl is at a salinity of 24. In the same samples we have found values over 1 µM for Fe (pre-*

*concentrated onto Nobias resin and analysed via HR-ICPMS, Krisch et al., in prep). Yes, the fluxes are overestimated due to the influence of Al rich river waters. We have a manuscript in preparation regarding Al in the Congo River plume and in Congo River waters and comparing it with other major world rivers.*

Al composition of dust – the D term in the equation: The 8% value that Measures and Brown used in the original MADCOW was mentioned in the text but I could not find anywhere what value the authors decided to use (should be around 2.69 mmol/g-1 if 8% Al by weight and 26.981539 is the molecular weight for Al) and if they varied this according to region. If it is constant then the term could be incorporated into the S term to reduce the model variables. How valid is the assumption that it is constant? Could not some of the variability in the S term be related therefore to variation in the D term if other studies made the same assumption? At the very least the value used should be included somewhere in the text. Some explanation of how this was handled in the current work would be most illuminating!

*Sorry. Our mistake. We have used the same value as in the original MADCOW model (8.1%). We do not have varied this value. It is used as a constant value. Our S term varies. We think is better to keep the D term in the model equation.*

*It has been postulated that the content of Al in dust is nearly invariant. The minimal differences on this value would not introduce a significant error on the calculations. Some of the variability in the S term could be related to variations in D. However, the impact of those variations in the present work is negligible.*

Specific Comments:

P4 line 20: (sp) The chemical reagent is known as Lumogallion, not Lumogallium.

*Corrected*

P4 line 33. For consistency the dissolved Al concentrations should be in µmol m-3 .

*Corrected*

P5 line 1. There is no explanation of what value is used for the D term in the equation. The other terms are explained in sections 2.2.1 – 2.2.3 but not the D term. If it is constant it could be included then in the S term.

*Indeed it is a constant. We have used the same value as in the original MADCOW manuscript.*

P5 line 7. This is a very large value Δσθ = 0.125 kg m-3 to use for determining the mixed layer depth as more recent work have shown that using smaller constraints Δσθ = 0.03 coupled with ΔT = 0.2° C provides a better estimate (de Boyer Montégut et al., 2004), this is in fact the threshold that is used in the Argo mixed layer climatology as cited in Holte et al. (2017). Thus it would be beneficial if the same criteria was used for the observed mixed layer depths to have a consistent approach. The problem with using a value Δσθ = 0.125 kg m-3 is that can seriously overestimate the mixed layer depth in high latitude areas leading to an increased inventory and longer residence time.

*Indeed you are right. However, since we did not use our own dAl inventories to derive the residence times of dAl over the Atlantic Ocean we therefore feel that the differences of using one threshold or the other one is not important. In this study, we averaged the dAl values found within the mixed layer.*

 P5 line 24. The authors should also be aware of work modelling the fractional solubility of aerosol Al (Han et al., 2012). It would therefore be prudent to include this work in the discussion and compare to the field data of Baker et al. (2013).

*We are aware of this study. However, we did prefer to use solubility estimates from field samples as there is a "good coverage" for the Atlantic Ocean. The estimates given by Han et al., 2012 also suffer from no measurements on the relation between the Al detachment rate or dissolution rate and pH. Also, Al solubility data used in the latter study are very scarce and only available from some cruises. We now mentioned the study of Han et al 2012 within the fractional solubility of Al section (2.2.2).*

P6 line 2. The residence time is a key variable in the version of MADCOW employed in this work and so it should be fairly well constrained. As the authors note the original version of MADCOW had the residence time fixed at 5 years along with the fractional solubility at 8% in order to simplify the calculations as changing one would impact the other. In the current approach it should be noted that the Han et al. (2008) work also includes many of the works that were not included in the citation list (see the general comments above) and these works were used to inform the residence times. It is also worth pointing out to the reader that Han et al. (2008) used a fixed mixed layer depth of 50 m and a constant solubility of 5% so this needs to be directly stated in the current manuscript with regard to how the values might compare.

*We have pointed out the reader the fixed mixed layer depth and the constant solubility*

P6 line 10. The modelled residence times will include advection and mixing to an extent, but the use of a fixed solubility and mixed layer depth will also induce some key differences for the regions examined in the present work. This likely explains why the residence times are longer in the Han et al. (2008) work than in others as for many locations, the underestimation of the solubility and the overestimation of the mixed layer will both work to increase the estimated residence time.

*You are right. In fact, some regions will show longer residence times and other regions shorter residence times in comparison with other studies. However, we needed to choose a benchmark in order to start our interpretations. We have mentioned the issues explicitly in the manuscript now. See end of section 2.2.3*

P7 line 9. See the general comment above regarding this extremely high value of dissolved Al.

*Indeed, this is a large value. We have answered to this comment within your general comment. However, it is averaged since more than one sample was taken within the mixed layer. We do have higher values up to 1.7 µmol of dAl (Menzel Barraqueta et al., in prep.). We also have measured dFe values over 1 µmol in the same waters (Krisch et al., in prep.).*

P8 line 12. There is a considerable amount of surface data for this region and compiling it all in one place may reveal more about the seasonal timings of the dust flux to this region and the aluminium response. See the general comment above regards other works that have data for this region.

*Indeed, there is a considerable amount of surface Al data for this region. However, many reported data are for unfiltered samples which does not match with our filtered dAl samples. We have added a number of additional references with comparisons of dAl data.*

P8 line 21. Not all of these studies attribute it to wet deposition, as the ITCZ acts partially as a barrier to the transport of the dust so the highest values are typically associated with direct dust deposition (Ravelo-Pérez et al., 2016; Tsamalis et al., 2013). Though precipitation is enhanced along the boundary between the ITCZ and the Saharan air layer (SAL) (Wilcox et al., 2010).

*We do not say that all authors attributed it to wet deposition. Our dAl maximum in the region coincide with minimum salinity values which is an indication of freshwater inputs.*

P9 line 2. From where does the Al rich upwelled waters come from? Al profiles normally decrease with depth (scavenged profile) so this needs to be explained further as it would have then be more likely to be resuspension of Al rich particles close to the shelf rather than a direct upwelling source.

*Certainly, the view that dAl concentrations normally decrease with depth is not uniformly correct. With all the new GEOTRACES data being published, it is clear that the dAl depth profiles are highly variable and that the distribution can resemble a scavenged type element but also a nutrient type element.*

*You are partially right. The sentence you point out comes from data presented in Bowie et al., 2002. There is not further discussion on type of source apart from coming from deeper waters. The same as upwelling of deep waters can induce phytoplankton blooms due to the supply of macronutrients and micronutrients it can also supply Al. We have not attributed this Al comes to either remineralization of biogenic particles or resuspension of sediments. However, both options could be correct and would have as a definite result the upwelling of Al rich waters.*

P9 line 3. Do you mean an increased number of particles or that they were enhanced in some other fashion? Larger? More sticky?

*Yes. Increased number of particles. We have reformulated the sentence to avoid confusion.*

P9 line 4. See the general comment on this above.

*Answered above*

P9 line 8. (sp) reported

*Corrected*

P10 line 2. A strong control of the fractional solubility is the relative humidity/hygroscopicity of the particle as this controls the pH, aerosol acidity (Keene et al., 2002).

*We have included this reference. "1) chemical processing during atmospheric transport which is influenced by the relative humidity of the particle (Keene et al., 2002), the balance of acid species (enhanced by anthropogenic sources e.g. fossil fuel combustion; (Ito, 2015; Sholkovitz et al., 2012) and the phase partitioning of $NH_3$ (Hennigan et al., 2015)*

P10 lines 22 and 24. This isn't a calculated result though, it is an estimate from a comparison with the work of Baker and colleagues.

*We have changed the word calculated for estimated*

P10 line 27. See the general comment about relating the fractional solubility to the precipitation or relative humidity levels in the atmosphere for these regions.

*We have replied to this issue above. We now explicit mentioned this issues in the text.*

P11 line 8. It would be useful to see a plot of the residence times (as a 2D map or property-property plot) to see how they look on spatial scales and in relation to primary productivity if it is the main loss term for Al in the mixed layer.

*We acknowledge your comment. However, we are not able to track down the residence time files from Han et al., 2008 in order to extract the actual modelled residence time for each station. Therefore we used a fixed value for each biogeochemical province.*

P11 line 33. It should be pointed out that statistically there are no differences between the values estimated here and those by Mahowald et al. (2005). So speculation on why the Mahowald is over estimated is somewhat spurious.

*This is not fully correct. There are regions were differences are statistically different and regions were they are not. See table 1 (first version of the manuscript).*

P13 line 25. The more northerly flux values are likely underestimated as the residence time used is too long as it is likely in reality, days to weeks (see discussion about this above). This is an important point as the MADCOW model should work well where the Al fluxes and concentrations are the highest.

*Indeed, you are right. We have added a couple of sentences showing that re adjusting the residence time for this region would yield much higher atmospheric fluxes which would make MADCOW calculated fluxes fit within previously reported atmospheric fluxes for this region.*

*See page 15 lines 24 to 27*

P14 line 2. Most likely – it clearly overestimates the fluxes when most of the Al is from the river.

*We have removed "most likely"*

P14 line 12. See the general comment above about relating the seasonality of the dust fluxes.

*Yes. However, this is really difficult to disentangle. The highly productive waters of the BENG region probably have a lower residence time of dAl during the upwelling season and probably a higher residence time during non-upwelling season. It is clear that if we would have sampled during low productivity season we may have found higher dAl concentrations as we found at the time of sampling. The latter would have yielded higher calculated dust fluxes. However, it has been acknowledged before the difficulties of the MADCOW to calculate "accurate" atmospheric fluxes in near-coastal regions and in such highly dynamic regions. Several cruises sampling and catching the seasonal variability of dAl in these waters would provide a better estimation of the fluxes.*

*We now have explicit mentioned this issues in the text.*

P14 line 25. So how do samples collected in the Pacific and Indian oceans tell us anything about deposition to the South Atlantic? Please explain this sentence more clearly.

*Wagener and colleagues performed model simulations to revise atmospheric deposition to the Southern Ocean. Their aerosol samples were taken in the Pacific Ocean and South of the Kerguelen Islands (Indian Ocean). However, they modelled the atmospheric deposition also for the South Atlantic. Therefore, they acknowledge that their largest uncertainties corresponded to regions downwind South America. The latter uncertainties arise from not taking into account or not having samples affected by Patagonian dust.*

*To avoid confusion we have remove the words 'from this sector'.*

P15 line 12. …lack of an island site…

*Corrected. " ..lack of island sites"*

P15 line 16. It is great that Al is measured on GEOTRACES cruises but this does not make this approach using MADCOW any stronger as the majority of the development of this type of work was done pre-GEOTRACES.

*We have modified the sentence. "Dissolved Al is a key trace element of the GEOTRACES programme and as such it is measured on all the GEOTRACES cruises which implies a great chance to use the MADCOW model"*

P15 line 21. Which IDP 2014 or 2017 – both are citeable now.

*The new one. 2017. We have added the reference.*

P15 line 23. For the Atlantic there are a number of north-south transects for Al and so some sort of seasonal signal is probably already possible and should be examined in the current work.

*You are right. We have acknowledge this in the text and use some historical data to decipher the influence of seasonality.*

*However, for a correct interpretation on the seasonal variability samples would need to be taken for the same transects and same stations (assuming the parcel of water is the same which is not the case). In this regard, many of the north to south transects sample different locations (similar in terms of the biogeochemistry) and such the seasonal signal could be bias by different conditions occurring at different stations.*

Figure 2: Please state in the caption the climatology range used here, is it over an annual cycle?

*Yes, it is the annual cycle. Caption corrected*

Table S5: The residence times used in this study are significantly shorter than what has been used previously in the MADCOW model (see above) and they are now on the same time scale as seasonal phytoplankton turnover so does this mean the residence time for Al can be scaled to productivity rather than input fluxes?

*It could be probably done if we consider that productivity is a measured of removal flux. However, you would need to add a non-biogenic removal term too. This would fit into the definition of residence time being the ratio of the dAl inventory in the mixed layer to the rate of input or removal. However, this is beyond the scope of this manuscript.*

Figure S4: The figure and the legend for this figure don't match up and there is no explanation of what the circles represent. While it is easy enough to conclude that the circles may represent discrete measurements at stations, the contoured data isn't explained and clearly does not share the same colour scale as the circles as the lowest value on the colour scale is blue and there is no blue in the contoured data. This figure needs to be fixed and explained better prior to acceptance.

*Done*